**Investigation**

# Linking molecular mechanisms to their evolutionary consequences: a primer

Rok Grah,[1] Calin C. Guet,[1] Gasper Tkačik,[1,*,†] Mato Lagator 🆔 [2,*,†]

[1]Institute of Science and Technology Austria, Klosterneuburg AT-3400, Austria
[2]Division of Evolution, Infection and Genomic Sciences, School of Biological Sciences, Faculty of Biology, Medicine and Health, University of Manchester, Manchester M13 9PL, UK

*Corresponding author: Institute of Science and Technology Austria, Am Campus 1, Lower Austria, Klosterneuburg, Austria, AT3400. Email: gasper.tkacik@ist.ac.at;
*Corresponding author: Division of Evolution, Infection and Genomic Sciences, Faculty of Biology, Medicine and Health, University of Manchester, Michael Smith Building, Manchester, UK, M13 9PT. Email: mato.lagator@manchester.ac.uk
[†]These authors contributed equally.

A major obstacle to predictive understanding of evolution stems from the complexity of biological systems, which prevents detailed characterization of key evolutionary properties. Here, we highlight some of the major sources of complexity that arise when relating molecular mechanisms to their evolutionary consequences and ask whether accounting for every mechanistic detail is important to accurately predict evolutionary outcomes. To do this, we developed a mechanistic model of a bacterial promoter regulated by 2 proteins, allowing us to connect any promoter genotype to 6 phenotypes that capture the dynamics of gene expression following an environmental switch. Accounting for the mechanisms that govern how this system works enabled us to provide an in-depth picture of how regulated bacterial promoters might evolve. More importantly, we used the model to explore which factors that contribute to the complexity of this system are essential for understanding its evolution, and which can be simplified without information loss. We found that several key evolutionary properties—the distribution of phenotypic and fitness effects of mutations, the evolutionary trajectories during selection for regulation—can be accurately captured without accounting for all, or even most, parameters of the system. Our findings point to the need for a mechanistic approach to studying evolution, as it enables tackling biological complexity and in doing so improves the ability to predict evolutionary outcomes.

Keywords: evolution; biological complexity; mechanistic model; gene expression

## Introduction

A major challenge in studying evolution is understanding how changes at the molecular level scale up and affect phenotypes, organismal fitness, and evolutionary outcomes (Doebeli *et al.* 2017; de Visser *et al.* 2018)—a challenge that is difficult to meet due to the complexity of biological systems. Some of the major factors that give rise to complexity in evolution include: (1) Mechanisms that determine the *effects of mutations* and the *epistatic interactions* between them. In other words, how mutations (genotype) alter the functioning of a system (phenotype), and how those changes to phenotype alter organismal fitness (Charlesworth and Charlesworth 2017; Yi and Dean 2019). While genotype-phenotype-fitness mapping has been extensively studied (Eyre-Walker and Keightley 2007; Pigliucci 2010; Lehner 2013; Kemble *et al.* 2019), it is either limited to the simplest systems or is not comprehensive (Szathmáry 1993; Schuster *et al.* 1994; Haldane *et al.* 2014; Yi and Dean 2019). (2) The intractably large mutational sequence space, which is impossible to explore experimentally and even computationally. Hence, a key challenge in predicting evolutionary outcomes is discerning how that space is *constrained*—understanding what phenotypes are at all accessible through mutations, and how those constraints arise from molecular mechanisms governing the

function of a system (Jacob 1977). Doing so minimizes the size of the mutational space that must be explored to accurately describe evolutionary dynamics. (3) The relationship between random sequences and their associated phenotypes, which is key for explaining de novo *evolution* of biological systems (Hledík *et al.* 2022). Understanding what phenotypes arise from random sequences further increases the sequence space that needs to be covered by genotype-phenotype-fitness mapping (Schuster *et al.* 1994; Yona *et al.* 2018; De Boer *et al.* 2020; Lagator *et al.* 2022). (4) The fact that the *distribution of mutational effects can change as the sequence evolves* (Orr 2003; Sanjuan *et al.* 2004; Couce *et al.* 2024), further raising the need for an accurate and comprehensive genotype-phenotype-fitness map that minimizes unjustified approximations. (5) The complex manner in which organisms navigate fitness landscapes, posing a difficulty in predicting what mutants, with *what phenotypes (and with what fitness) get fixed in the population* as a consequence of selection (de Visser and Krug 2014; Lässig *et al.* 2017).

To illustrate these various factors that make the evolutionary process complex, consider the study of bacterial gene regulation and its evolution. For the simplest components of gene regulation in prokaryotes—constitutive promoters that only bind RNA polymerase (RNAP)—predictive genotype-phenotype-fitness mapping at steady-state expression levels is available (Kinney *et al.*

2010; Einav and Phillips 2019; Lagator *et al.* 2022). In other words, we can predict the *effects of mutations* in constitutive promoters and even, to an extent, the *epistatic interactions* between them (Lagator, Paixão, *et al.* 2017). Furthermore, there is a solid mechanistic understanding of the relationship between random sequences and their associated phenotypes, which allows not only predictions of gene expression levels from random sequences but also of de novo *promoter evolution* (Payne and Wagner 2014; Tuğrul *et al.* 2015; Aguilar-Rodríguez *et al.* 2018; Lagator *et al.* 2022). The existing understanding of GP mapping in constitutive prokaryotic promoters lends itself to describing the structure of promoter fitness landscapes as well as the phenotypic outcomes of selection in bacteria (Otwinowski and Nemenman 2013; Haldane *et al.* 2014; Aguilar-Rodríguez *et al.* 2017), yeast (De Boer *et al.* 2020; Vaishnav *et al.* 2022), and multicellular organisms (Duque *et al.* 2014; Fuqua *et al.* 2020). Put together, we have a solid basis to study how the steady-state expression levels arising from constitutive promoters in prokaryotes can evolve, as we can predict how any mutation alters the phenotype and, to a lesser extent, how those phenotypic changes might impact organismal fitness.

However, even gene regulation at a single promoter is notably more complex than just steady-state expression levels based on RNAP binding alone. Most prokaryotic promoters are regulated by at least 1 transcription factor (TF), which can either reduce (repressor) or enhance (activator) the binding of RNAP. The fact that promoters are regulated, as opposed to constitutively expressed, means that they can exist in multiple states corresponding to each of the environments they are responsive to. In addition, following environmental change that affects the concentration of the TF in the cell, regulated promoters go through a transient phase before reaching a new steady-state expression level (Longo and Hasty 2006; Yosef and Regev 2011; Shih and Fay 2021). This means that the promoter sequence, as well as the TF and RNAP concentrations, affects not only steady-state phenotypes (one expression level per environment) but also the multiple phenotypes describing the dynamics of gene expression.

The main aim of this work is to provide a primer on how to study biological systems of increasing complexity in order to understand their evolution. Building on previous works that adopted a mechanistic approach to study the evolution of gene regulation (Ackers *et al.* 1982; Dean and Thornton 2007; Vilar 2010; Josephides and Moses 2011; Pai *et al.* 2015), we argue that knowing the mechanisms governing system's function can be utilized to ask detailed questions about its evolution and to simplify the complexity of the system without sacrificing accuracy of evolutionary predictions.

To this end, we developed a novel modeling approach that captures the dynamics of gene expression from a bacterial promoter regulated by RNAP and a repressor. Focusing on the dynamics of gene expression, as opposed to only the steady-state expression levels, allowed us to connect a genotype (promoter sequence) to 6 phenotypes which might all contribute to overall fitness. While answering some specific questions about the evolution of this system, our aim was to present a primer on how to utilize interdisciplinary approaches to connect the mechanistic, molecular level understanding of a system to the key quantities that define its evolution and how doing so allows accounting for greater complexity. To this end, we considered the most complete and complex version of our model, which accounted for the largest number of parameters describing how the system works, as the benchmark against which we compared simpler model versions to understand how complexity could be reduced. Each section of this article explores the utility of our mechanistic approach when trying to understand a key evolutionary property: (1) the *phenotypic effects of mutations*; (2) the nature of *constraints* in GP mapping; (3) de novo *evolution* of sequences toward new functions; (4) how the *distribution of mutational effects depends on current fitness*; and (5) what *phenotypes (and with what fitness) get fixed in the population*.

# Materials and methods
## Model
### Experimental system and measurements
We used a synthetic system based on the Lambda phage switch, in which we decoupled the *cis-* (promoter) and *trans-* (transcription factor) regulatory elements, as previously described in Lagator, Paixão, et al. (2017). We removed *cI* and substituted *cro* with *venus-yfp* (Nagai *et al.* 2002) under control of $P_R$ promoter, followed by a T1 terminator sequence. The $O_{R3}$ site was removed in order to remove the $P_{RM}$ promoter. Separated by a terminator sequence and 500 random base pairs (bp), we placed *cI* under the control of $P_{TET}$, an inducible promoter regulated by TetR (Lutz and Bujard 1997), followed by a TL17 terminator sequence. In this way, concentration of CI TF in the cell was under external control, achieved by addition of the inducer anhydrotetracycline (aTc). The entire cassette was inserted into a low-copy number plasmid backbone pZS* carrying a kanamycin resistance gene (Lutz and Bujard 1997).

We measured the ON->OFF dynamics of gene expression in the wild-type $P_R$ system in the following manner. Six replicates were grown overnight in M9 media, supplemented with 0.1% casamino acids, 0.2% glucose, and 50 µg/ml kanamycin. The absence of the inducer aTc indicates that these cells were grown in the ON state overnight. Overnight cultures were diluted 100×, grown for 2 h under the same conditions, and then diluted again at 100×. At this point, each replicate population was diluted into 2 conditions: same as the overnight growth (in this case, ON state); different state to the overnight, in this case achieved by adding 10 ng/ml aTc. Fluorescence of growing replicate populations was measured every 10 min in Bio-Tek Synergy H1 platereader. The measured fluorescence was always corrected for the autofluorescence of the media. Populations were always grown at 37°C. To measure OFF->ON dynamics, we used the same protocol, but have grown overnight cells in the presence of 10 ng/ml aTc. These wild-type $P_R$ measurements served as the basis to derive model parameters.

### Thermodynamic model
The thermodynamic model is a well-established model for gene regulation which provides a highly quantitative mapping from promoter sequences to gene expression levels that is compatible with biophysical measurements (Shea and Ackers 1985; Bintu, Buchler, Garcia, Gerland, Hwa, Kondev, Kuhlman, *et al.* 2005; Bintu, Buchler, Garcia, Gerland, Hwa, Kondev, and Phillips 2005; Kinney *et al.* 2010; Lagator *et al.* 2022). It uses statistical mechanics to describe equilibrium probabilities of different molecules binding to the sequence of interest, and uses these to model the expression levels of the gene of interest.

The thermodynamic model requires us to know: (1) all the possible binding configurations; (2) binding energies (and interacting energies) associated with each binding configuration; and (3) intracellular concentrations of the binding molecules.

*Binding configurations.* Binding configurations are specific to each system—in our system, the following binding states are possible (Fig. 2b):

1) empty state, i.e. nothing is bound;
2) RNAP bound to, e.g. $P_R$ promoter;

3) one CI dimer bound to either $O_{R1}$ or $O_{R2}$;

4) two CI dimers cooperatively bound to both $O_{R1}$ and $O_{R2}$.

In each of these binding states, different binding locations are possible. For example, RNAP can bind to its strongest binding site at −35 and −10, or at any other part of the sequence. Of course, binding to other parts of the promoter sequence is often very unlikely and will contribute very little to total binding. However, when there is no one clear strong binding site, binding to these weaker binding sites can cumulatively contribute to expression (Lagator et al. 2022).

Besides the 4 states accounted for by our model, there are other possible configuration, such as RNAP and CI both binding at the same time to different binding sites (without steric hindrance). Another example would be 3 CI dimers simultaneously bound to the DNA. However, these configurations are extremely unlikely and contribute negligible amount to total binding on any promoter sequences we used in this study. They would become significant (and important to include) only if strong RNAP and CI binding sites would not overlap. We include some of them in the evolutionary calculations—for details on that see Methods section Evolutionary model.

*Binding energies.* Each of the 4 configurations has an energy of binding that is obtained using EMs of RNAP and CI. The energy matrix (EM) contains the information about how every possible point mutation in the DNA binding site of a given molecule impacts its overall binding energy. As such, each DNA binding molecule has a unique EM associated with it, which can be thought of as a unique representation of that molecule's function, much like the amino acids sequence is a 2D representation of that molecule's 3D structure. Therefore, in our system, we require 2 EMs, 1 for description of RNAP and 1 for CI (Supplementary Fig. 1). EMs are $4 \times L$ matrices whose elements give the energy contribution of the given nucleotide (rows) at given position (columns) to the total binding energy. The total binding energy is then the linear sum of individual energies, summing up a contribution of each residue at each position in the binding site (Fig. 2b). We used the previously published EMs for RNAP (Lagator et al. 2022) and the extrapolated EM for Lambda CI (Igler et al. 2018).

*Expression and probability of expressing state.* One of the main assumptions of the thermodynamic model is that the rate of expression—and thus steady-state expression level—is proportional to the probability that expressing state occurs. In our system, the expressing state is the one that contains RNAP bound to the promoter. Therefore, we can write the probability of finding the system in the state with RNAP bound as follows:

$$P_E = \frac{\Sigma_i [\text{RNAP}] e^{-E_i^R}}{1 + \Sigma_i [\text{RNAP}] e^{-E_i^R} + \Sigma_i [\text{CI}_2] e^{-E_i^{CI}} + \Sigma_i [\text{CI}_2]^2 e^{-E_i^{CI} - E_{i+24}^{CI} + \varepsilon}} \quad (1)$$

The numerator is the Boltzmann weight of the RNAP bound state, while the denominator represents the sum of Boltzmann weights of all possible configurations. $E_i^R$ and $E_i^{CI}$ are binding energies of RNAP and CI, respectively, to binding site i, which represents every possible binding position along the sequence. [RNAP] and [$CI_2$] represent the available RNAP and CI dimer concentration, respectively, and $\epsilon > 0$ represents the cooperativity energy between 2 CI dimers whose binding sites are 24 bp apart (distance normally observed in the Lambda $P_R$ promoter). For the wild-type $P_R$ sequence, there is only 1 significant RNAP binding site, and 2 CI binding sites

($O_{R1}$ and $O_{R2}$) to which CI can cooperatively bind. All energies are in the units of $k_BT$.

*The relation between CI monomer and dimer concentrations.* As binding to $O_{R1}$ and $O_{R2}$ occurs by CI dimers (quantity required in the thermodynamic model) and not monomers [quantity obtained from the mass action kinetics (MAK) model], we compute the relationship between the two. Let us denote the rate of 2 CI monomers forming a CI dimer as $k_1$ and the opposite dissociation rate as $k_2$. We can assume that the system is in chemical equilibrium, meaning that the processes of dimerization and dissociation occur faster than the changes in CI concentration. The relationship between CI monomer and dimer concentrations can therefore be expressed as: $2[\text{CI}] \leftrightarrow_{k_2}^{k_1} [\text{CI}_2]$ (note that $k_1/k_2$ marks the rates of forward and reverse reactions, respectively).

This means that we can rewrite eq. (1) as:

$$P_E = \frac{\Sigma_i [\text{RNAP}] e^{-E_i^R}}{1 + \Sigma_i [\text{RNAP}] e^{-E_i^R} + \Sigma_i \omega_1 [\text{CI}]^2 e^{-E_i^{CI}} + \Sigma_i \omega_1 [\text{CI}_2]^4 e^{-E_i^{CI} - E_{i+24}^{CI} + \varepsilon}} \quad (2)$$

where $\omega_1 = k_1/k_2$ contains the rates describing relation between CI monomers and dimers.

*Reference points of energies.* The quantities appearing in Boltzmann weights (eq. 2) are the binding energies of RNAP and CI and their concentrations. The binding energies are relative to the unbound state. However, the EM produces only the change in binding energy, relative to the reference point—the sequence of the wild-type Lambda $P_R$ promoter.

Given that the binding energies in eq. (2) are the changes relative to these reference points, the binding energies of the reference points must be taken into account:

$$P_E = \frac{\Sigma_i g_1 [\text{RNAP}] e^{-E_i^R}}{1 + \Sigma_i g_1 [\text{RNAP}] e^{-E_i^R} + \Sigma_i \omega [\text{CI}]^2 e^{-E_i^{CI}} + \Sigma_i \omega [\text{CI}_2]^4 e^{-E_i^{CI} - E_{i+24}^{CI} + \varepsilon}} \quad (3)$$

where $g_1 = e^{-E_{wt}^R}$ represents the Boltzmann weight of RNAP binding to the wild-type $P_R$ sequence, and $\omega = \omega_1 e^{-E_{wt}^{CI}}$ is the combination of dimer/monomer rates ($\omega_1$) and Boltzmann weights of CI binding to the wild-type sequence of $O_{R1}$. Note that, as $E^R$ and $E^{CI}$ represent the energies relative to the reference points (wild-type $P_R$ sequence), their binding energies to the reference sequence are by definition zero.

*Changes in CI concentration are much slower than the equilibration of the system.* The thermodynamic model described above gives the prediction of expression where all quantities are assumed to be in equilibrium. However, in our system, the concentration of repressor varies, potentially violating this assumption. Yet, if the time scales on which CI concentration varies is much slower than the time scale on which the equilibrium is established, the assumption of equilibrium would still be satisfied. In our case, the time scale of varying CI concentration is hours, much longer than the rates at which RNAP transcribes DNA.

## MAK model

The second part of the model is the MAK, which follows the concentration of repressor CI and fluorescence protein YFP. The concentration of CI is used to model the probability of RNAP being bound and, hence, the rate of *yfp* expression (eq. 3), while the YFP concentration is used as a proxy for gene expression. We used 2 Ordinary Differential Equations (ODEs), 1 for each concentration. Both have 2 terms, one that describes the production of

the molecule, and the second with processes that lower the concentration.

We model CI concentration as:

$$\frac{d[CI]}{dt} = R_{CI}f_{CI}(t) - \frac{[CI]}{\tau_{CI}} \tag{4}$$

where $R_{CI}$ represents the production rate of CI, $f_{CI}(t)$ is the delay in production rate that takes values between 0 and 1, and $\tau_{CI}$ describes the effects of dilution and degradation.

When studying the dynamics of gene expression in the system switching from an ON to OFF state, the $P_{TET}$ promoter is induced, leading to expression and production of CI. However, there exist a delay between the introduction of the inducer into the system and the CI repressor binding to the promoter. This delay is due to (1) diffusion of inducer into the cell and to its binding target, TetR repressor; (2) transcription and folding of CI protein; and (3) diffusion of CI to its cognate binding site. We do not model the details of these 3 contributions in details but lump them into 1 delay, described by:

$$f_{CI}(t) = \frac{t^n}{t^n + \tau_1^n} \tag{5}$$

where $\tau_1$ is the effective time scale of delay, and $n$ is the effective Hill coefficient (or sharpness) of delay. This equation ensures that the production rate of CI for $t \ll \tau_1$ is zero, while for $t \gg \tau_1$, production rate converges toward $R_{CI}$.

Similarly, the production of CI for the OFF->ON dynamics is delayed by the inducer of the system being removed from the system. This occurs due to the inducer (that allows CI production by binding to TetR and removing it from the $P_{TET}$ promoter) lingers in cells after they are transferred into the ON environment (no CI and hence no inducer). This leads to continued, but diminishing, production of CI in the environment where no inducer is present, which reaches the steady state when the inducer is diluted through cell division as the inducer is not actively removed from cells. We describe this process by:

$$f_{CI}(t) = \frac{\tau_2^\beta}{t^\beta + \tau_2^\beta} \tag{6}$$

where $\tau_2$ is the effective time scale, and $\beta$ is the effective Hill coefficient (or sharpness). This equation ensures that for $t \ll \tau_2$, the production of CI is $R_{CI}$, from which point it decreases toward zero when $t \gg \tau_2$.

Finally, we rely on the above to model the concentration of YFP, which is in our study used as a proxy for gene expression levels:

$$\frac{d[YFP]}{dt} = R_{YFP}P_E(t) = \frac{[YFP]}{\tau_{YFP}} \tag{7}$$

where $R_{YFP}$ represents the basal production rate of YFP, and $P_E(t)$ is the probability of RNAP being bound (leading to expression), obtained from eq. (3). $P_E$ changes as a function of time as the concentration of CI (which appears in $P_E$) also changes with time. $P_E$ takes values between 0 and 1. Here, we assume that the probability of RNAP being bound is linearly proportional to the rate of expression, as typical for the thermodynamic model of gene expression (Bintu, Buchler, Garcia, Gerland, Hwa, Kondev, and Phillips 2005). $\tau_{YFP}$ describes the dilution and degradation of the YFP protein.

## Obtaining the parameters for the model

Each of the models (TD and MAK) require a different set of parameters. The TD model includes the following parameters (eq. 3): (1) prefactor in the RNAP bound state $g_1[RNAP]$ (which we can treat as 1 parameter); (2) scaling factors that determine the units of EM elements $\alpha$ and $\iota$ for RNAP and CI energy matrices, respectively; (3) prefactor in CI bound states, $\omega$; (4) cooperativity $\epsilon$ between 2 CI dimers bound at $O_{R1}$ and $O_{R2}$. The MAK model parameters include (eqs. 4 and 7): (1) YFP and CI production rates ($R_{YFP}$ and $R_{CI}$, respectively); (2) dilution and degradation times of YFP and CI ($\tau_{YFP}$ and $\tau_{CI}$, respectively); and (3) parameters $\beta$, $n$, $\tau_1$, and $\tau_2$ describing delay in the production of CI. These model parameters were obtained from different independent datasets, described below.

*Thermodynamic model parameters.* The parameters for the thermodynamic model are the parameters that describe the steady-state expression—the expression in ON and OFF states without any temporal dynamics between the states. These parameters were obtained from previously published works (Igler *et al.* 2018; Lagator *et al.* 2022): $g_1[RNAP] = 3.27$; RNAP EM scaling $\alpha = 4.85\ k_BT$; cooperativity between CI dimers $\epsilon = 3.22\ k_BT$; $\omega[CI]_{steady\ state}^2 = 0.01$; and CI EM scale $\iota = 3.00\ k_BT$. The EMs were also obtained from these published works (Supplementary Fig. 1).

*MAK model parameters.* The MAK model parameters are those that describe the temporal change in CI and YFP concentration. The steady-state concentration of CI is given as follows:

$$\frac{d[CI]}{dt} = R_{CI}f_{CI}(t) - \frac{[CI]}{\tau_{CI}} = 0 \tag{8}$$

$$[CI]_{steady\ state} = R_{CI}f_{CI}(t)\tau_{CI} = \begin{cases} 0, & \text{in ON environment} \\ R_{CI}\ \tau_{CI}, & \text{in OFF environment} \end{cases} \tag{9}$$

Alternatively, the steady-state concentration of YFP is expressed as follows:

$$\frac{d[YFP]}{dt} = R_{YFP}P_E(t) - \frac{[YFP]}{\tau_{YFP}} = 0 \tag{10}$$

$$[YFP]_{steady\ state} = R_{YFP}P_E\tau_{YFP}$$

$$= \begin{cases} R_{YFP}P_E^{ON}\tau_{YFP}, & \text{in ON environment} \\ R_{YFP}P_E^{OFF}\tau_{YFP}, & \text{in OFF environment} \end{cases} \tag{11}$$

where $P_E^{ON}$ and $P_E^{OFF}$ represent $P_E$ in ON and OFF states, respectively. In $P_E$, the appropriate value of $[CI]_{steady\ state}$ is used.

*YFP production rate only determines units of YFP.* We next show that YFP production rate, $R_{YFP}$, only determines the units of YFP concentration. In other words, YFP production rate only scales with YFP concentration. If we rewrite $[YFP] = [yfp] \cdot R_{YFP}$, and use this in eq. (7), we obtain an ODE with rescaled YFP concentration (marked by [yfp]) but where the production rate does not appear in the ODE:

$$R_{YFP}\frac{d[yfp]}{dt} = R_{YFP}P_E(t) - R_{YFP}\frac{[yfp]}{\tau_{YFP}} \rightarrow \frac{d[yfp]}{dt} = P_E(t) - \frac{[yfp]}{\tau_{YFP}} \tag{12}$$

As YFP production rate is not present in the ODE, it follows that it does not determine the dynamics of YFP in our model.

*CI production rate is determined from the steady-state OFF expression.* Similarly as for YFP, CI production rate also determines only the units of CI concentration and not its dynamics.

As the maximum effective steady-state concentration of CI is already determined by the steady-state expression in the presence of CI, we set $R_{CI}$ to be such that the constraint $\omega[CI]_{steady\ state} = 0.01$ is met. In practice, this means we can set $\omega = 0.01$ and $[CI]_{steady\ state} = 1$, following that $R_{CI} = 1/\tau_{CI}$.

*Normalization of YFP to the wild-type ON expression.* Due to experimental reasons, YFP expression values in experimental data have arbitrary units. Therefore, we decided to normalize all YFP expression measurements and predictions by wild-type $P_R$ ON expression. In other words, the wild-type ON expression is set to have value 1. The steady-state concentration of YFP of the wild-type $P_R$ promoter in the ON environment is therefore written as:

$$[YFP]_{WT}^{ON} = R_{YFP}\tau_{YFP}P_E^{WT}([CI] = 0) \tag{13}$$

$$= R_{YFP}\tau_{YFP}\frac{\Sigma_i g_1[RNAP]e^{-E_{i,WT}^R}}{1 + \Sigma_i g_1[RNAP]e^{-E_{i,WT}^R}} \tag{14}$$

Effectively, we set $R_{YFP} = 1$, and normalize all YFP results by $[YFP]_{WT}^{ON}$. One can think of this expression as constraining the YFP production rate $R_{YFP}$ such that wild-type expression in ON environment equals 1.

*Determining the dynamical MAK parameters.* The remaining parameters that needed to be determined were $\tau_{CI}$ and $\tau_{YFP}$, which capture the dilution and degradation rate of CI and YFP, respectively; and the 4 parameters that describe delay in CI production ($\tau_1, \tau_2, n$, and $\beta$). We used the wild-type temporal expression curves, from ON->OFF and OFF->ON to fit these parameters. We obtained that $\tau_{CI} = \tau_{YFP} = 60$ min, which corresponds to the dilution of both molecules due to cell growth. This agrees with the fact that active degradation of both YFP and CI is generally very slow in cells. We obtained $\tau_1 = 70$ min; $\tau_2 = 70$ min; $n = 2$; and $\beta = 5$ which are all within the expected range of values.

*Model agreement with the data*

To validate the performance of the model, we created 9 $P_R$ promoter mutants, which our model predicted to affect the binding of RNAP and CI in different ways: (1) not to significantly affect the binding of either; (2) primarily impair RNAP binding; (3) primarily impair CI binding; and (4) impair the binding of both, RNAP and CI (Supplementary Table 1). Importantly, all the parameters were obtained from either independent steady-state datasets (Igler *et al.* 2018; Lagator *et al.* 2022), or from the expression dynamics of the wild-type $P_R$ system only. This means that the prediction of these mutants is parameter free as no parameter was fitted from this set of data.

We ordered oligonucleotides containing the desired mutants from Sigma-Aldrich, and cloned them into the wild-type $P_R$ system by restriction/digestion. We verified each cloned mutant by Sanger sequencing. We measured the dynamics of their expression in the same manner as that of the wild-type $P_R$ promoter (see section Experimental system and measurements).

To test the goodness of fit, we computed the Pearson correlation coefficient between all the time points of the 9 predicted and measured temporal dynamics of gene expression. We obtained that $\rho_{ON\to OFF} = 0.90$ and $\rho_{OFF\to ON} = 0.90$. Next, we tested the predictive power of the MAK model independently, as TD models have been evaluated previously on this (Igler *et al.* 2018; Lagator *et al.* 2022) and similar systems (Vilar 2010; Razo-Mejia *et al.* 2014). To test only the goodness of fit of the MAK model,

we wanted to remove the potential error in determining the steady-state expressions. In other words, if the steady-state values are wrong, this will result in the wrong prediction of the temporal dynamics. Therefore, we normalized all the temporal dynamics curves of both model and experimental data in such a way that they all shared the same starting and ending point. This way, we compared if the model and data trajectories that now share the same ON and OFF steady-state expression levels, also share the same dynamics (Fig. 2; Supplementary Fig. 2). The agreement between model and data is very high with Pearson correlation coefficient $\rho_{ON\to OFF} = 0.98$ and $\rho_{OFF\to ON} = 0.96$. These evaluations quantitatively show that, while TD gives good prediction and represents the state-of-the-art modeling, the MAK model gives almost perfect prediction of the dynamics with very little deviations from the experimental data (Fig. 2).

## Calculation of phenotypic landscapes

To compute the phenotypic landscapes, we use all double mutants of the wild-type sequence. The reason why we have not used single mutants is that for the sequence of length $L = 67$ bp, there are only 201 single mutants, most of them having little or no effect on the phenotypes (Supplementary Fig. 3). In contrast, the total number of double mutants is $\approx 20,000$, which gives high enough sample to explore the properties of the phenotypic landscapes without being too computationally demanding.

## Phenotypic landscapes with continuous energies

To obtain the binding energies to a given binding site, we use the EM on the binding site sequence. However, to disentangle the effects of the discreteness of EMs and the specific promoter architecture (i.e. the relative position of binding sites in the promoter), we explored the phenotypic landscapes where binding energies are not constraint by the sequence but can take any value in a continuous. This gave us the limits to the phenotypic space that the system can explore imposed by biophysical constraints, excluding the effects coming from the sequence (discrete energy and architecture of binding sites). Even though there are potentially many binding sites for RNAP and CI, we took into account only the bindings to the strongest binding sites: RNAP binding to its strongest binding site on the promoter and CI binding to $O_{R1}$ and/or $O_{R2}$. In other words, we assume only these 3 binding sites exist which is a valid approximation as binding to other positions is much less likely. This gave us a 3D problem where 3 binding energies are independently and continuously varied.

Furthermore, to fairly compare the double mutant phenotypic landscapes of continuous energies with energies determined from the EMs, we limited the range of continuous binding energies. The range of the binding energies to the 3 sites (1 for RNAP and 2 for CI) was limited to the range that can be explored by double mutants when modeled using the discrete energies from the EMs. For example, for continuous RNAP binding energy, we find (using the EM) the highest and lowest energy of binding to $P_R$ for double mutants. This range is between $-5.32$ and $9.70$ $k_BT$ (where wild-type binding energy is the reference point with energy zero). The range of binding energies for $O_{R1}$ and $O_{R2}$ is $(-0.75, 7.43)$ $k_BT$ and $(-1.69, 7.43)$ $k_BT$, respectively.

## Varying overlap between $O_{R1}$ and $-10$ region

To study the effects of overlap between RNAP and CI binding site, we varied the overlap between $O_{R1}$ and $-10$ region of the wild-type $P_R$ sequence. To do this, we rely on the fact that an EM is a representation of the corresponding molecule's binding function. This meant that we could adjust the RNAP EM in such a way

that the −10 region was moved, changing the number of bp that overlap between −10 and $O_{R1}$. Specifically, if the −10 position was moved by $h$ bp downstream, we increased the spacer between −35 and −10 by $h$. The spacer penalties were also corrected, allowing now for $h$ larger spacer. By moving the −10 position, we have effectively changed the size of EM to $4 \times (L + h)$. To keep the binding to wild-type sequence unaffected by this change, we adjusted the wild-type entries within the EM. As per our definition, the EM elements representing wild-type sequence have a value zero. This means that in each EM column, there is one element with zero energy contribution, representing wild-type nucleotide (Supplementary Fig. 1). The 3 remaining elements have nonzero values and represent the energetic effects of mutating from the wild-type to another nucleotide in that position. Therefore, each column in the new EM was adjusted, such that the element representing WT sequence was assigned zero value, while the remaining 3 elements in the column were given the corresponding 3 nonzero values. In other words, the wild-type nucleotide in the column was adjusted due to the movement of −10 region by $h$ bp. This ensured that expression of the wild-type sequence was always predicted to be the same, for all tested promoter architectures.

## Shuffling the elements of the EMs

We randomly shuffled the elements of an EM without repetitions. This maintained the original distribution of elements in the EM but destroyed any internal structure of the EM. To have a common reference point between different EMs, we fixed the elements in the EM that represent the wild-type sequence. This way, the expression of the wild-type sequence was not affected by the shuffle.

## Computing the surface area of phenotypic landscapes

A set of points in space does not have a volume (or surface). Therefore, to compute the surface area of a phenotypic landscape, we assumed that each mutant is represented with a square of edge length $a$, which is centered around the point of the mutant in phenotypic space. Assuming a square instead of a circle is computationally easier to implement, as surface area of a set of non-disjoint circles is a nontrivial task. To compute the surface area of a set of mutants in the phenotypic space, we first represented the phenotypic space with a grid. The size of each tile in the grid was much smaller than the size of mutant's square $a$. Next, for each mutant, we marked which tiles of the grid are covered by this mutant's square. Doing this for all mutants allowed us to calculate the surface area of mutants in the phenotypic space, which was proportional to the total number of marked tiles.

We measured all 6 phenotypes first for the wild-type promoter (with the exception of OFF expression which was in units of wild-type ON expression), meaning that the scale of each phenotypic value was 1. Therefore, our square size for each mutant was set to be $a = 0.01$, representing a small change in phenotypes which could potentially be explored by intrinsic noise. However, the surface areas of various phenotypic landscapes do not qualitatively change with other values of $a$ (Supplementary Fig. 8).

## Evolutionary model

To model how random sequences evolve regulated promoter function, we used the strong-selection-weak-mutation (SSWM) model with single point mutations being introduced into the system (Gillespie 1983; Tuğrul *et al.* 2015). The model assumes that point mutations are rare such that at any given time only a single mutation is competing to be fixed in the population. The next

mutation emerges only when that mutation is fixed or eliminated from the population. As the average time scale is determined by the arrival of a new mutation, we use the inverse mutation rate as the unit of time for estimating how quickly promoters evolve.

## What configurations lead to expression/repression

For the wild-type system, there already exist strong RNAP binding sites with 2 CI binding sites overlapping it. However, when evolving a regulated promoter de novo, it is not entirely clear which promoter architectures (relative location of RNAP and CI binding sites) are productive. In the absence of more detailed experimental knowledge of how promoter architecture impacts promoter regulation, we assumed that the binding of CI downstream of the RNAP binding sites or if their binding sites overlapped in sequences would lead to repression. We considered binding of CI upstream of the RNAP binding sites not to lead to repression by itself (although, note, that it could contribute to repression by stabilizing binding to another CI binding site through cooperativity).

## Fitness function

To describe the fitness function around its global maximum/ peak, we used the Taylor expansion around it and wrote the quadratic term. However, as we did not want to consider situations when fitness was negative, as we started from random sequences that did not have promoter function and assumed that no improvement in their function would have a negative impact on the host. This assumption is not necessarily accurate, as, for example, constitutive expression without repression could indeed negatively affect organismal fitness. These more complex scenarios were beyond the scope of this work. Hence, we described fitness as

$$ F \approx 1 - \frac{s}{d} \sum_{i=1}^{d} \left( \frac{p_i}{p_i^*} - p_i^{\mathrm{opt}} \right)^2 \tag{15} $$

where $s$ represents the selection coefficient, the sum goes over different phenotypes $i$, and $p_i^*$ is the normalizing factor of phenotype $p_i$ that determines the units. $p_i^* = p_i^{\mathrm{WT}}$ for all phenotypes, except for OFF expression which is measured in units of ON wild-type expression, i.e. $p_{\mathrm{ON}}^{\mathrm{WT}}$, as by construction. $p_i^{\mathrm{opt}}$ is the optimal value of $p_i$ with highest fitness and equals $p_i^{\mathrm{opt}} = 1$ for all phenotypes except for OFF expression, which has value of $p_i^{\mathrm{opt}} = 0.0012$. In other words, we expressed phenotypes in their wild-type units (with the exception of OFF expression), and modeled expression toward wild-type $P_R$ values of each phenotype.

The sum inside the fitness function can be written as:

$$ \sum_{i=1}^{d} \left( \frac{p_i}{p_i^*} - p_i^{\mathrm{opt}} \right) = \overbrace{(\mathrm{ON} - 1)^2 + (\mathrm{OFF} - 0.0012)^2}^{2D} + \left( \frac{\mathrm{slope}_{\mathrm{ON \to OFF}}}{\mathrm{slope}_{\mathrm{ON \to OFF}}^{\mathrm{WT}}} - 1 \right)^2 $$
$$ + \left( \frac{\mathrm{slope}_{\mathrm{OFF \to ON}}}{\mathrm{slope}_{\mathrm{OFF \to ON}}^{\mathrm{WT}}} - 1 \right)^2 + \left( \frac{\mathrm{lag}_{\mathrm{ON \to OFF}}}{\mathrm{lag}_{\mathrm{ON \to OFF}}^{\mathrm{WT}}} - 1 \right)^2 + \left( \frac{\mathrm{lag}_{\mathrm{OFF \to ON}}}{\mathrm{lag}_{\mathrm{OFF \to ON}}^{\mathrm{WT}}} - 1 \right)^2 \tag{16} $$

where ON and OFF are already, by construction, in the units of wild-type ON expression. If evolving only steady-state expression, only the first 2 parts of the sum are taken (marked by brackets with 2D). Alternatively, evolving all 6 phenotypes requires all 6 contributions. This makes sure that the optimal

value of fitness is $F = 1$, while $F < 1$ for any other nonoptimal phenotypes.

However, the fitness function in eq. (15) is a quadratic form, approximating the peak only around the neighborhood of the peak. To ensure that the fitness function is limited between 0 and 1 throughout the entire landscape, we generalized the fitness function to

$$F = \exp\left[-s\frac{1}{d}\sum_{i=1}^{d}\left(\frac{p_i}{p_i^*} - p_i^{\mathrm{opt}}\right)^2\right] \tag{17}$$

which can be approximated by a quadratic form for phenotypes close to the optimal value.

### Fixation probability

The fixation probability is given by the Kimura fixation probability

$$p_{\mathrm{fix}} = \frac{1 - e^{-2\Delta F}}{1 - e^{-2\Delta FN}} \tag{18}$$

where $N$ is the population size and $\Delta F = \frac{F_{\mathrm{new}}}{F_{\mathrm{old}}} - 1$, or the relative change in fitness between the old and the new mutation. As typical bacterial population sizes are relatively large, the denominator will mostly have a small contribution.

**Thermodynamic model applied to random sequences.** To compute the probability of expressing state, we must take into account those binding configurations that were highly unlikely when we were exploring only the mutational neighborhood of the wild-type $P_R$ sequence. This means that we need to extend the set of the binding configurations that we model to include: (1) unbound state; (2) only RNAP bound; (3) only CI bound; (4) CI bound upstream of RNAP; and (5) CI bound downstream of RNAP. As mentioned above, the productive states that lead to expression are (2) and (4). We can write the probability of states (2) or (4) occurring as:

$$p_E = \frac{w_2 + w_4}{1 + w_2 + w_3 + w_4 + w_5} \tag{19}$$

$$w_1 = 1 \tag{20}$$

$$w_2 \propto \sum_{i=1}^{M-L_R+1}[\mathrm{RNAP}]e^{-E_i^{\mathrm{CI}}} \tag{21}$$

$$w_3 \propto \sum_{i=1}^{M-L_{\mathrm{CI}}+1}[\mathrm{CI}]^2 e^{-E_i^{\mathrm{CI}}} \tag{22}$$

$$w_4 \propto \sum_{i=1}^{M-L_{\mathrm{CI}}-L_R+1}[\mathrm{CI}]^2 e^{-E_i^{\mathrm{CI}}}\sum_{i=1}^{M-L_R+1}[\mathrm{RNAP}]e^{-E_i^{\mathrm{R}}} \tag{23}$$

$$w_5 \propto \sum_{i=1}^{M-L_R-L_{\mathrm{CI}}+1}[\mathrm{RNAP}]e^{-E_i^{\mathrm{R}}}\sum_{i=1}^{M-L_{\mathrm{CI}}+1}[\mathrm{CI}]^2 e^{-E_i^{\mathrm{CI}}} \tag{24}$$

where $w_{1-5}$ represent Boltzmann weights for states (1)–(5); $M$, $L_R$, and $L_{\mathrm{CI}}$ are the total sequence length, the RNAP binding site length, and the CI binding site length, respectively. All other model formulations were the same as described above.

### Computation and visualization of phenotypic and fitness landscape

If we rewrite the Boltzmann weights as $w_3 = [\mathrm{CI}]^2\overline{w_3}$, $w_4 = [\mathrm{CI}]^2\overline{w_4}$, and $w_5 = [\mathrm{CI}]^2\overline{w_5}$, we can write the probability of expression as:

$$p_E = \frac{w_2 + [\mathrm{CI}]^2\bar{w}_4}{1 + w_2 + [\mathrm{CI}]^2(\bar{w}_3 + \bar{w}_4 + \bar{w}_5)} \tag{25}$$

which can be represented as

$$p_E = \frac{K_1 + [\mathrm{CI}]^2 K_2}{1 + K_1 + [\mathrm{CI}]^2(K_2 + K_3)} \tag{26}$$

where $K_1 = w_1$ represents productive configurations with only RNAP bind; $K_2 = \overline{w_4}$ configuration with CI bound upstream of RNAP, both leading to expression. $K_3 = \overline{w_3} + \overline{w_5}$ represents unproductive configurations that do not leading to expression—see eq. (19).

For a fixed value of $K_1$, $K_2$, $K_3$, all phenotypes are exactly determined via TD and MAK models. Therefore, using these 3 parameters, we can characterize the whole fitness landscape (shown in Fig. 6g).

### Geometric model

To test how the description of the evolutionary dynamics of evolving promoters depended on accounting for the underlying biophysical mechanisms of promoter function, we used 2 alternative models. In the first, the phenotypic values of mutations were drawn from a geometric distribution ("Geometric model on phenotype"). Here, mutations had an effect on phenotypes, as opposed to modeling their effect on Boltzmann weights first. The second considered that mutations alter binding energies (and through them Boltzmann weights) first, and through those changes affected the phenotypes ("Geometric model on binding energies").

### Geometric model on phenotypes

In the geometric model where the effects of each mutation were reflected in a random change in the phenotypes, we varied all phenotypes in the following way: each mutation had a fixed effect size in the phenotypic space, meaning that the mutant had a fixed Euclidean distance in the phenotypic space from the initial point. This meant that vector of changes in phenotypes was described by $\vec{dp} = \Sigma_i dp_i$, where $dp_i$ is a change in phenotype $i$ and $|\vec{dp}|$ is fixed. Therefore, the new value of phenotype $i$ is $p_{\mathrm{initial}} + dp_i$, where $p_{\mathrm{initial}}$ is the initial value of this phenotype. To obtain $\vec{dp}$, we randomly drew numbers in the range of $(-1, 1)$ for each of the phenotypes, and at the end normalizing the vector to the desired amplitude. We used $\vec{dp} = 0.3$.

### Geometric model on Boltzmann weights

As we can represent our phenotypic landscape with 3 effective Boltzmann weights (see section Evolutionary model), we also modeled each mutation as an effective change in the sizes of Boltzmann weights. Similarly as for phenotypes, the effects of mutations were represented by a vector $\vec{dp} = \Sigma_i dr_i$ of fixed size in the Boltzmann weight space. However, as Boltzmann weights take values that vary many orders of magnitude, we decided that each mutation will have a relative effect on each Boltzmann weight: $K_i^{\mathrm{mutant}} = K_i(1 + dr_i)$, where $K_i$ represents Boltzmann weight $i$. We constrain $\vec{dp}$ to be fixed at 1. Relative change in Boltzmann weights can be represented by an additive change in effective binding

energies of the 3 productive configurations of the system. We can write each of the 3 Boltzmann weights as $K_i = e^{-E_i}$, $i \in \{1,2,3\}$, with $E_i$ representing the effective binding energy of configuration $i$. Therefore, as each mutation leads to a relative change in Boltzmann weights $K_i$, we can write $K_i^{mutant} = K_i(1 + dr_i)$ and $-E_i^{mutant} = \log K_i^{mutant} = \log (K_i(1 + dr_i)) = \log K_i + \log dr_i = -E_i + dr_i$, if $dr_i \ll 1$.

### Computing distribution of fitness effects for the geometric model

To compute the distribution of fitness effects (DFEs) of the 2 geometric models, we used the original model (Fig. 2) and randomly drew sequences (to avoid any bias) until we found a genotype with the desired fitness value. This was done to determine the original sequences into which we introduced every possible single and double mutation to obtain the DFE, as we did with the original model (Fig. 3). Having obtained the Boltzmann weights and phenotypes of this genotype from the original model, we computed all possible single mutants around it using either the Geometric model on phenotypes or on effective Boltzmann weights. For each mutant, we randomly drew a vector of fixed size $\overrightarrow{dr_1}$, which represents the change in either phenotypes or Boltzmann weights. To compare the DFEs of these 2 models with the original model, we also needed to compute double mutant effects. We did this by applying the procedure described above twice; first to obtain single mutants, and the second time to get double mutants. In other words, first a random vector $\overrightarrow{dr_1}$ was applied on the original phenotypes/Boltzmann weights to model the effects of single mutations, and then, a second random vector $\overrightarrow{dr_2}$ was applied on the new phenotypes/Boltzmann weights of the single mutation. This provided a double mutant effect. To compute the means and standard deviations of these DFEs, we used 30 sequences with different genotypes for each given fitness value (Fig. 6d).

### Obtaining the architecture of Escherichia coli promoters

Using the data from RegulonDB, we obtained sequences of all known *Escherichia coli* promoters regulated by $\sigma^{70}$ sigma factor (~2,000) and the consensus binding sequences of all known *E. coli* TFs obtained based on ~3,700 TF binding sites from *E. coli* (Tierrafría *et al.* 2022). To find the cognate binding sites for each TF, we matched the consensus sequences of each TF to each of the ~2,000 promoters, considering a TF to be involved in the regulation of a promoter if its consensus binding sequence can be found within that promoter sequence. To find the −35 and −10 region of each promoter, we used the $\sigma^{70}$–RNAP EM (Lagator *et al.* 2022) and considered the position with highest likelihood (lowest binding energy) for $\sigma^{70}$–RNAP to be the primary binding site. We focused only on the $\sigma^{70}$ promoters as it is the only sigma factor with a known EM. Furthermore, to be consistent with our experimental system, we used only the data where the TF was known to have the repressor function, leading to the mapping of 700 TF binding site positions relative to the RNAP binding site. These were then classified in one of the 3 architectures (Fig. 7a).

## Results
### Effect of mutations on multidimensional phenotypes
#### Experimental system

In order to develop a mechanistic model that can predict the effects of mutations on dynamical gene expression phenotypes, we focused on the canonical model system in bacterial genetics—the Lambda bacteriophage promoter $P_R$ (Ptashne *et al.* 1980). Relatively simple regulated promoters, such as Lambda $P_R$, are the fundamental building blocks of gene regulatory networks and, hence, provide a relevant starting point for understanding the forces that shape gene regulatory evolution. $P_R$ is a repressible promoter: in the absence of the TF CI, RNAP binds to the −10 and −35 sites with high affinity and leads to strong expression; when CI is present in the system, it cooperatively binds to its 2 binding sites, $O_{R1}$ and $O_{R2}$, competitively preventing RNAP from binding, and in doing so repressing the promoter (Fig. 1b). In the experimental synthetic system we used, we placed the *cI* gene under an inducible $P_{TET}$ promoter on the same small copy number plasmid—SC101* origin with 2–3 copies (Lutz and Bujard 1997)—as the $P_R$ promoter, enabling external control of CI concentrations. The $P_R$ promoter, which controlled the expression of a yellow fluorescence marker (*yfp*) in our system, was also modified to exclude the $O_{R3}$ site, and, with it, the $P_{RM}$ promoter that is typically present on the reverse complement. This choice simplified the regulation of the system while also ensuring expression does not occur on the opposite strand. The plasmid was placed in the MG1655 K12 strain of *E. coli*, modified to express the tetracycline repressor (TetR).

In our experiments, this system could exist in 2 distinct states (Fig. 1). In the "ON" state, CI is not present, and hence, only RNAP binding determines YFP expression levels. In the "OFF" state, CI is present at a high concentration, fully repressing the wild-type $P_R$ promoter. In order to study the dynamics of gene expression in this system, we considered switching in both directions: from "ON->OFF" state and from "OFF->ON" state. In other words, we would maintain the system under one condition (either "ON" or "OFF") for a sufficiently long time to ensure that steady-state expression levels are reached. Then, we induce the other state by either stimulating ("ON->OFF") or stopping ("OFF->ON") *cI* expression. Note that we consider dynamics of gene expression at the population rather than single cell level, meaning that dynamics in our work are the average response over all the individuals in the population. We summarized the steady-state and the dynamics of gene expression through 6 distinct phenotypes (Fig. 1b): (1) steady-state "ON" expression level; (2) steady-state "OFF" expression level; (3) the duration of the lag when the system is switching from "ON->OFF", defined as the time from induction of the system to the point when the expression level is 50% of the maximum range between "ON" to "OFF"; (4) lag when the system is switching from "OFF->ON"; (5) the slope at the point lag is measured when the system is switching from "ON->OFF"; and (6) the slope when the system is switching "OFF->ON". We also sometimes considered the amplitude (the difference between "ON" and "OFF" expression levels), but we did not treat it as a distinct phenotype. To avoid the obvious effect of amplitude on the slope (twice the amplitude would mean twice the slope), we rescaled the slope with the amplitude. While we define 6 phenotypes, an increase compared with the more common approach of observing only steady-state expression levels, it is still a simplified way of describing dynamics of gene expression.

### Effect of mutations on multidimensional phenotypes—predictive model

To describe the steady-state and temporal dynamics of gene expression, we combined 2 modeling approaches—the statistical thermodynamic model of steady-state expression and the MAK (Fig. 2).

The thermodynamic (TD) model describes the mapping between the genotype and the steady-state expression levels (Shea and Ackers 1985; Bintu, Buchler, Garcia, Gerland, Hwa, Kondev,

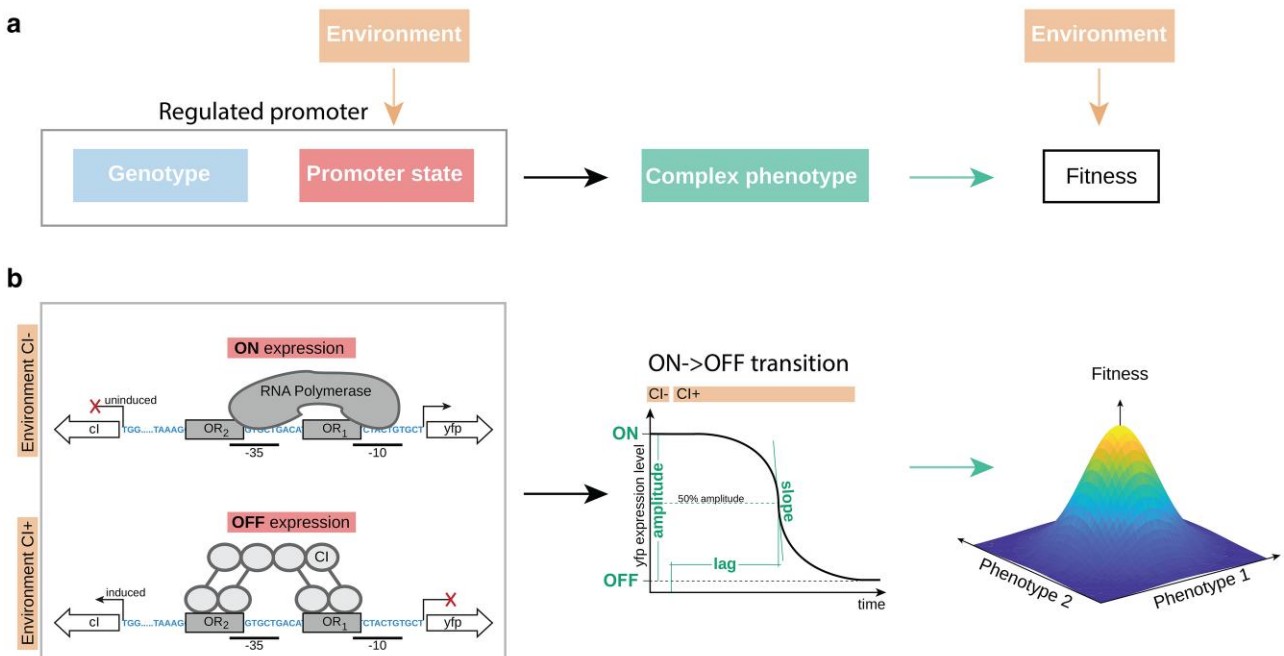

**Fig. 1.** Regulated promoter as a model system to study evolution of systems with complex phenotypes. a) A regulated promoter can be characterized through its genotype (blue) and a specific promoter state (ON, OFF, or the transient, dynamical, state between them, red). The promoter state is determined by the environment (orange). Hence, a change in the environment gives rise to a complex set of phenotypes that capture the steady-state and the dynamics of gene expression (teal) and which, in turn, alter the fitness of the organism in an environment-dependent manner. b) We used components of the Lambda bacteriophage $P_R$ promoter as the model regulated promoter. Specifically, the promoter, controlling the expression of a yellow fluorescent marker protein *venus-yfp*, contains a strong RNAP binding site (marked as −10 and −35) and 2 repressor (Lambda CI) binding sites ($O_{R1}$ and $O_{R2}$). On the opposite strand of the same small copy number plasmid, we placed the Lambda *cI* gene under an inducible control of the $P_{TET}$ promoter. The environment exists in 2 possible states, defined by the presence (OFF expression) or the absence (ON expression) of the inducer of the repressor CI. The switch between these 2 states leads to a dynamical change in the gene expression phenotype, either from ON->OFF (shown here) or OFF->ON, from which we measure several phenotypes—amplitude, slope, and lag. The system can be described through 6 phenotypes (2 dynamic phenotypes for ON->OFF, 2 dynamic for OFF->ON dynamics, and 2 steady-state phenotypes). The changes in these phenotypes depend on the promoter sequence. Changes to phenotypes alter fitness, with optimal fitness in this study defined as the value of the 6 phenotypes corresponding to that of the wild-type Lambda $P_R$ promoter. Note that a different definition of the optimal fitness would not qualitatively alter any of the presented findings.

and Phillips 2005). For a given promoter and the molecules that bind and regulate it, the TD model calculates the equilibrium probability of finding the system in all possible binding states and then assumes that the steady-state expression levels are proportional to the probability of finding the system in a productive state. Specifically, the model system used in this study (Fig. 2) can be in 4 distinct states (Fig. 2b, part II): (1) no molecules bound to the promoter; (2) RNAP bound; (3) repressor bound; and (4) 2 repressors bound cooperatively to both binding sites. From these 4 states, only (2) is productive and leads to transcription. To calculate the energy of binding for each molecule that binds the $P_R$ promoter, the TD model uses the EMs of RNAP and CI (Supplementary Fig. 1), as well as the strength of cooperative binding between 2 repressors (relevant only in state 4). The EM contains the information about how every possible point mutation in the DNA binding site of a given molecule impacts its overall binding energy (Kinney *et al.* 2010; Lagator *et al.* 2022). We obtained the EMs for RNAP and CI (Supplementary Fig. 1), as well as all other parameters required to predict *steady-state* expression levels with the TD model, from published works (Sarai and Takeda 1989; Igler *et al.* 2018; Lagator *et al.* 2022).

The MAK uses standard ODEs to describe the temporal dynamics of molecular concentrations in a system. In our system, MAK accounts for the changes in concentrations of the repressor CI and the measurable system output, YFP. While we assume a constant and high concentration of RNAP, the concentrations of CI and YFP

change due to their variable production and dilution rates. MAK also models the delay in response of the regulatory elements to an external condition (in our system, the addition or removal of the inducer). MAK assumes the probability of binding from the TD model to be proportional to the rate of production of YFP (Fig. 2b, part III). To fit the MAK parameters, and hence, to model the *dynamics* of the system, we minimized the mean square error of the predicted vs measured temporal dynamics of the wild-type, unmutated $P_R$ system in both "ON->OFF" and "OFF->ON", discussed below.

To validate the performance of this model, we created 9 $P_R$ promoter mutants (Supplementary Table 1), predicted to affect the binding of RNAP and CI in qualitatively different ways: (1) not to significantly affect the binding of either; (2) primarily impair RNAP binding; (3) primarily impair CI binding; and d) impair the binding of both, RNAP and CI. These mutants were predicted based on the TD model and carried between 2 and 7 mutations each compared to the wild-type $P_R$ promoter. We measured the temporal dynamics of these mutants when switching from "ON->OFF" and "OFF->ON", and found that our combined model predicted their gene expression dynamics well (Fig. 2c and d; Supplementary Fig. 2). The bulk of the error in predictability of our model came from the TD part, likely due to the low accuracy of the CI EM [as we previously observed (Igler *et al.* 2018)]. We further confirmed that the error mostly originated from the TD part by fixing steady-state expression levels to the measured values (which were previously predicted using the model trained on

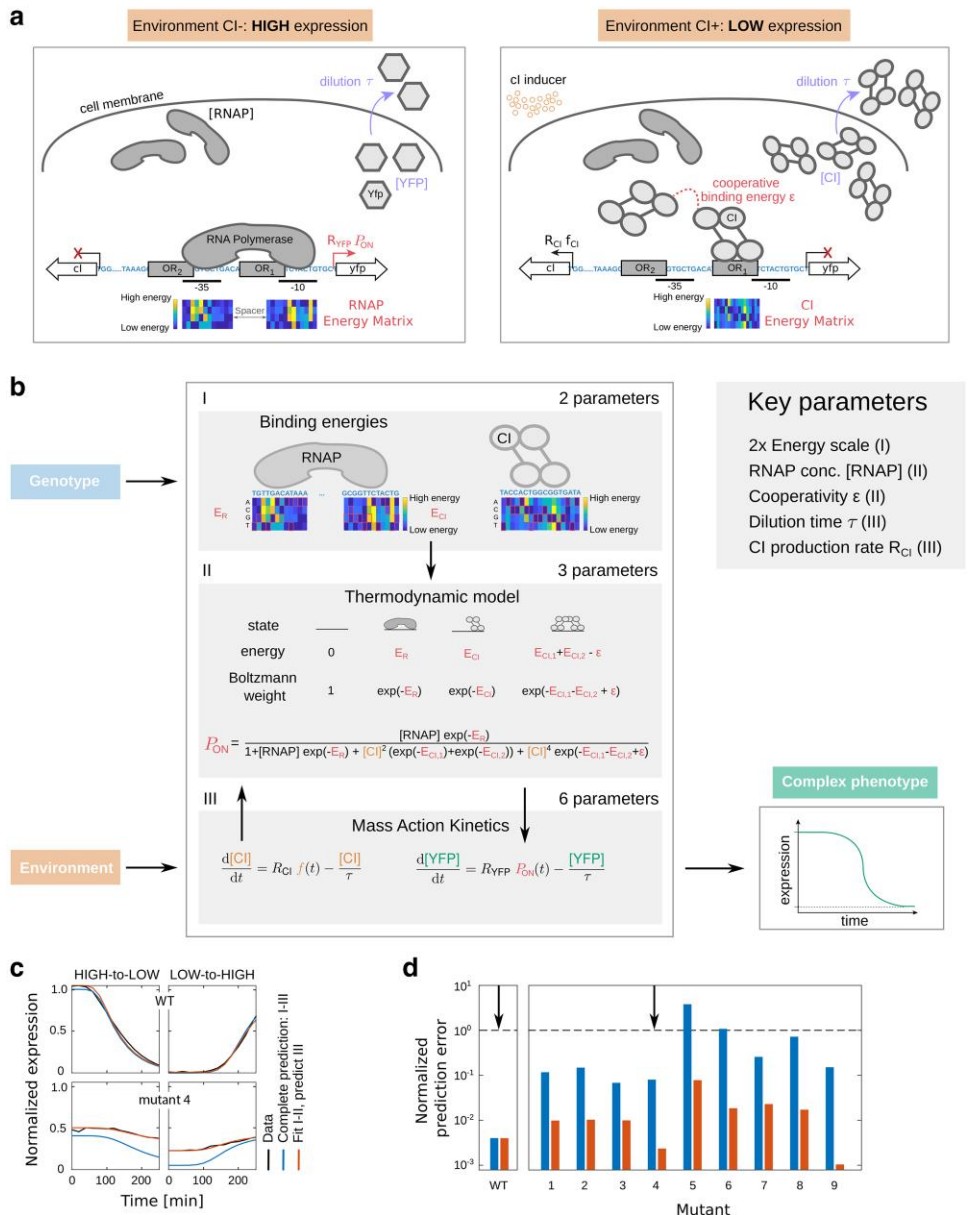

**Fig. 2.** Modeling multidimensional gene expression phenotypes. a) Schematic of the mechanisms involved in gene regulation that were explicitly modeled, shown for 2 environments—without (ON) and with (OFF) Lambda CI. b) Details of the model that link genotype to phenotype. RNAP EM is used to determine the binding energy of RNAP to a given sequence ($E_R$), which determines the ON expression levels. The ability of CI to repress, when present in the environment, depends on its binding energy to $O_{R1}$ ($E_{CI,1}$) and $O_{R2}$ ($E_{CI,1}$) (determined by its EM), as well as the cooperativity between 2 CI dimers ($\epsilon$). Red rectangles in the matrices indicate the lowest energy residue for each position in the binding site of RNAP or CI. The thermodynamic part of the model (I and II) predicts the probability of the promoter being in one of 4 possible states: empty (unbound), RNAP bound, 1 CI molecule bound, or both CI molecules bound cooperatively. It does so based on the binding energies of RNAP and CI to a given promoter sequence, and the concentrations of those molecules. Gene expression levels are then considered proportional to the probability of finding the system in the productive state ($P_{ON}$), when RNAP is bound. We assume the concentration of RNAP is constant in a cell. We use MAK to determine the concentration of the repressor based on its production rate ($R_{CI}$) and dilution rate stemming from cell division ($\tau$) (III). To model the expression level of the fluorescence marker, and hence the multidimensional phenotypes of the given promoter mutant, MAK part of the model relies on the calculated probability of RNAP being bound ($P_{ON}$) and the dilution rate ($\tau$). All model parameters were inferred only from the multidimensional gene expression phenotypes of the wild-type $P_R$ promoter, not any of the mutant promoters. c) Model predictions of multidimensional gene expression phenotypes for the wild-type $P_R$ promoter and one mutant (marked with arrows in d). Model predictions made by predicting every aspect of the models (I–III) were less accurate than the predictions made by fitting the thermodynamic component of the models (I and II) and predicting the MAK component (III). d) Prediction error for all 9 tested mutants, based on predicting all components of the model (blue) or those based on fitted thermodynamic components (red). Most of the error of the models come from the poor predictions of the thermodynamic component of the model, likely stemming from the limited precision of the CI EM.

the wild-type alone) and found the model fit dramatically improved temporal dynamics for all mutants (Fig. 2c and d).

While inferring a more precise model from a larger dataset, one that better accounts for other mechanisms involved in regulating gene expression levels, or a nondeterministic model that captures

the stochasticity in gene expression, would better reflect on the reality of bacterial gene regulation, that was not the main aim of our study. Rather, we looked for a model that captures gene expression dynamics sufficiently well to serve as a source of biophysically realistic effects of mutations, allowing us to tackle

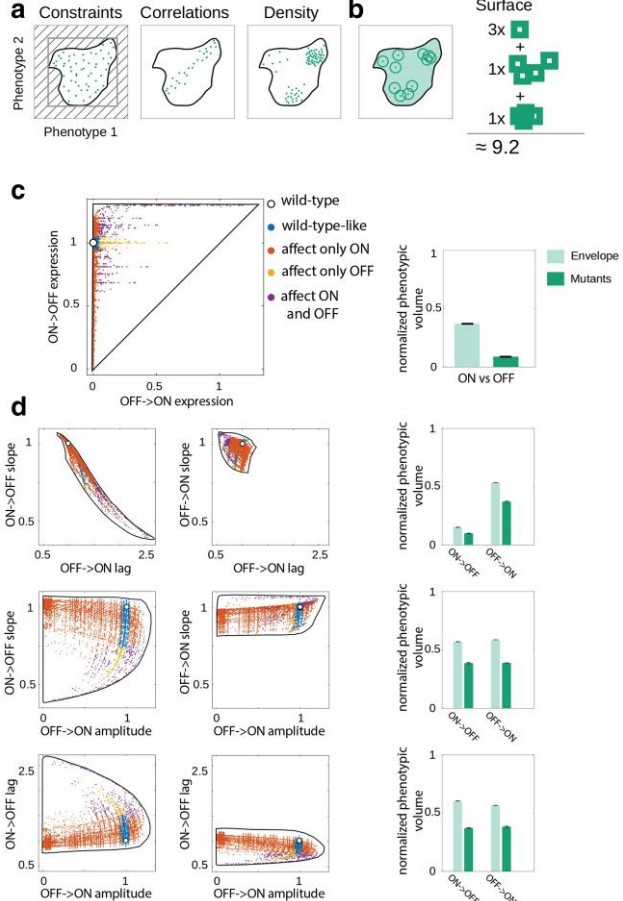

**Fig. 3.** Constraints on GP mapping. a) Illustration of the 2 ways in which we define constraints between 2 phenotypes: (1) the maximum area that is achievable through mutations (the *envelope*), shown as the thick black line; and (2) the *phenotypic surface* of the envelope covered by mutations, capturing the correlations between mutations and their density in the phenotypic space. b) The phenotypic surface is calculated by allowing each mutation to occupy a pre-defined rectangle around the value of each of its associated 6 phenotypes, and then calculating the surface area covered by those rectangles (for more information, see Computing the surface area of phenotypic landscapes). c) Left: phenotypic landscape of double mutants visualized in 2D, in this case between steady-state ON and OFF expression levels. The black line is the *envelope*, showing the total range of phenotypes that can be achieved through mutations. The white point shows the wild-type $P_R$ promoter phenotypic value; colored circles are the phenotypic effects of mutations. The color of points indicates how the mutation alters steady-state ON and OFF expression levels. Right: 2 ways of quantifying constraints, both calculated as normalized phenotypic volume: (1) the ratio of the envelope (defined as illustrated in a) and the bounding box that is defined by the extremal values of the envelope (gray rectangle in a), shown in light teal; and (2) the ratio of area of all double mutant volumes (defined as illustrated in b) and the bounding box that is defined by the extremal values of the envelope (gray box in a), shown in dark teal. d) Left: visualization; and Right: quantification of the constraints, for all possible pairs of phenotypes. Colors of points are the same as in c). Note that quantifying constraints 1 phenotype at the time (i.e. in 1D) would yield higher quantities for normalized phenotypic volume, while quantifying in 3 or more dimensions would produce lower estimates. We examined 2 phenotypes at the time for ease of visualization.

biological complexity in the form of multidimensional phenotypes. While still approximate, we therefore consider our most complex version of the model (i.e. containing most detail about the underlying mechanisms) to "accurately reflect promoter evolution" and benchmark less complex models against it.

## Constraints on GP mapping

Starting with the wild-type Lambda $P_R$ promoter, we used our model to exhaustively explore the effect of all possible single and double mutants on all 6 phenotypes (Fig. 3; Supplementary Fig. 3). We were specifically interested in the correlations between phenotypes, asking whether mutations alter the phenotypes independently of each other or not. We refer to these correlations as *constraints*, as they limit the possible states that the system can adopt and define what phenotypes can be achieved through mutation, or, in other words, the distribution of phenotypic effects of mutations. While the constraints can be understood as statistical dependencies in the full 6D space of measured phenotypes, we can conveniently visualize them in 2D projection planes that represent all possible pairs of phenotypes.

We measure the constraints acting on GP mapping by 2 quantities. First, we determine the *envelope*—the total range of phenotypes that could possibly emerge in the system (Fig. 3a). We do this by allowing the binding energies between RNAP/CI and the promoter to assume any continuous energy value from the minimum to the maximum energy value observed in the EM, rather than the discrete energy penalties accessible via point mutations. The envelope, therefore, gave us the limits to the phenotypic space that the system can explore arising from the biophysical constraints inherent to protein-DNA binding, excluding the effects arising from the specific TFs (their EMs) or the specific architecture of the promoter (the position of the TF binding sites). Second, we quantified how evenly the envelope surface is explored through mutations (Fig. 3b). To get a more complete picture of the constraints shaping GP mapping, we focused on double mutants (Fig. 3), as they cover the 2D phenotypic space more fully than the single mutants (Supplementary Fig. 3). Together, these 2 measures quantify the phenotypic space that could possibly be explored and how that space is actually explored through mutations.

The GP map of the $P_R$ promoter is heavily constrained, as double mutants explore only a portion of the possible landscape (Fig. 3c and d). For example, with respect to the 2D combination of "ON" and "OFF" phenotypes, only ~40% of the possible space can be explored by any number of mutations. Strikingly, double mutants can explore less than 10% of that space (Fig. 3c), indicating heavy constraints acting on steady-state expression levels. In fact, it is not possible to explore more than 55 or 40% of the total phenotypic space through any number of mutations or, specifically 2 mutations, respectively (Fig. 3d), meaning that a large portion of the possible phenotypic space (the envelope) can never be accessed through a smaller number of point mutations in the promoter.

The observed heavy constraints do not imply that the system is robust and that mutations cannot drastically alter one or more phenotypes. In fact, many double mutants have a large effect on the phenotypes. This finding goes against a common assumption of quantitative genetics—that small genetic changes (i.e. individual mutations) lead to small phenotypic changes (Milocco and Salazar Ciudad 2020). While the observed constraints do not imply that the system is robust, they do set a limit to the possible phenotypic states that can be achieved. A more constrained system is less likely to lead to evolutionary innovations (Ciliberti *et al.* 2007), as mutations result in a smaller set of possible phenotypic states, limiting the extent to which the system can explore the full, unconstrained phenotypic landscape. These constraints give rise to canalization in bacterial promoters (Wagner *et al.* 1997), whereby the same value of 1 phenotype can be achieved by many mutations. Because a more constrained system can assume a reduced number of possible phenotypic states, evolution is also more likely to be repeatable and to undergo the

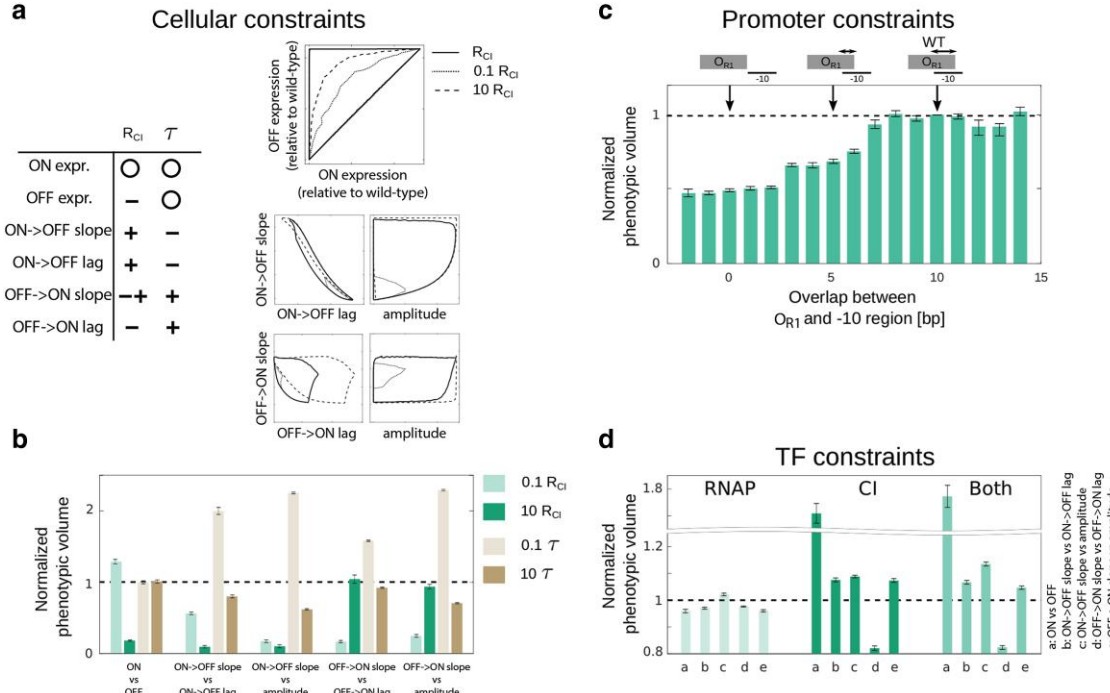

**Fig. 4.** Mechanistic origins of GP constraints. a) The role of 2 key parameters in the MAK—CI production rate ($R_{CI}$) and the dilution rate of CI and YFP ($\tau$)—on the phenotypic constraints. The table shows the effect of the 2 parameters on phenotypic constraints (here measured as the total size of the envelope), with "−" and "+" indicating a negative and a positive effect, respectively, "− +" a nonmonotonic relationship, and the circle indicating that phenotype is not affected by the parameter. The plots show visually how the envelope changes with the changes in CI production rate. b) Impact of CI production rate ($R_{CI}$) and the dilution rate of CI and YFP ($\tau$) on the normalized phenotypic volume, i.e. the extent to which mutations evenly explore the space within the envelope. c) The role of promoter architecture on the constraints (here defined as the normalized phenotypic volume). Promoter architecture is defined as the number of bp that overlap between the RNAP and CI binding sites. d) Role of RNAP and CI EM structures on the set of pairwise normalized phenotypic volumes, explored by shuffling the EMs. To do this, the distribution of EM elements was maintained, and only their position shuffled. Bars are mean phenotypic volumes of 1,000 randomized EMs (either RNAP, CI, or both EMs were randomized), normalized to the phenotypic volume of the wild-type matrices (dashed line).

same pathways during the adaptive process (de Visser and Krug 2014).

## Constraints on GP mapping—mechanistic origins

The model also allowed us to understand the mechanistic origins of the constraints observed in Fig. 3. Understanding not only *what* mutations do but also *why* is critical for developing a more predictive understanding of evolution, as it enables generalizing beyond a specific system being studied.

Several key properties of the system that might impact the constraints on GP mapping include: (1) the concentration of the TF; (2) the architecture of the promoter, meaning the relative position of TF and RNAP binding sites; and (3) the factors that impact the binding energies of RNAP and CI.

### TF concentrations

The concentration of the CI repressor in the $P_R$ system is affected by its production, and dilution rates. The production rate is determined by the inducer, while the dilution rate results from the combined effect of the cell division, transmembrane dilution, and protein degradation. In our model, the output of the system, YFP, has the same dilution rate as CI. Our model reveals that CI production and dilution rates impact most phenotypes individually and, in most cases, in a monotonic fashion (Fig. 4a; Supplementary Fig. 4). When considering the constrains that emerge between pairs of phenotypes, CI production and dilution rates alter the limits of phenotypic values that can be achieved (i.e. the envelope) as well as the manner in which mutations explore the envelope (Fig. 4a and b).

In other words, the concentrations of molecules in the system affect the maximum phenotypic range that can be achieved and how evenly that space is explored.

### Promoter architecture

Each bacterial promoter has a specific architecture, determined by the relative position of RNAP and TF binding sites in it. The wild-type $P_R$ promoter contains 1 strong RNAP binding site consisting of the −10 and −35 elements, and 2 binding sites for the CI repressor, $O_{R1}$ and $O_{R2}$. In the wild-type promoter, $O_{R1}$ has a 10 bp overlap with the extended −10 RNAP element [note that our recent work identified more than the core 6 bp region of the −10 element to directly affect RNAP binding (Lagator *et al.* 2022)]. This means that mutating those 10 positions in the promoter affects the binding of both RNAP and CI simultaneously. In order to more clearly understand the role that promoter architecture plays in constraining the 6 gene expression phenotypes, we considered a changing number of nucleotides that overlap between $O_{R1}$ and the extended −10 foot of RNAP. By construction, CI binds $O_{R1}$ in each architecture variant with equal affinity—an assumption that ignores the potential role of other, poorly understood mechanisms involved in TF-DNA binding (see Methods section Varying overlap between $O_{R1}$ and −10 region).

The critical property that changes as the overlap between the binding sites of 2 molecules changed was the number of positions that, when mutated, affect the binding of both instead of just 1 molecule. We expected that greater overlap would result in greater constraints, as the effects of mutations on RNAP and CI binding would be correlated. However, we instead found that smaller overlap led

to increased constraints (Fig. 4c), meaning that promoter architectures with more independent binding of RNAP and CI have a stronger correlation between phenotypes and hence could explore a smaller portion of the total phenotypic landscape surface area. This somewhat counter-intuitive finding stems from the fact that, when there is no overlap, a point mutation can affect either the binding of CI or of RNAP, while with overlap it can affect the binding of one, the other, or, critically, both simultaneously. In fact, binding site overlap can magnify the effects of point mutations in a system like ours. For example, one mutation can simultaneously increase RNAP binding and decrease CI binding leading to a greater increase in expression levels than can be achieved if the binding sites do not overlap. Such nontrivial interactions can only be elucidated through a mechanistic model.

### Binding energies

The fundamental summary of a key function of bacterial TFs (namely, their binding affinity to DNA) is contained within their EMs, which describe the effect of every possible point mutation in the binding site on the energy of binding between a given TF and DNA. Constraints in GP mapping can arise from 2 properties of each EM (Supplementary Fig. 1): some mutations within a position affect the binding energy more than others; and some positions contribute more to the overall binding than others.

We explored the extent to which the specific structure of RNAP and CI EMs affected the constraints in the multidimensional phenotypes. To do this, we created 1,000 alternate EMs for both, RNAP and CI, in which we kept the wild-type sequence intact (meaning that the sequence with the lowest binding energy was always the same) but shuffled randomly the specific entries in the matrix.

For most pairs of phenotypes, shuffled RNAP EMs decreased, while shuffled CI EMs increased, the total surface area explored by mutations (Fig. 4d). In other words, the wild-type RNAP imposes fewer constraints than one would predict based on randomized EMs, while the wild-type CI imposes greater constraints. This might be a consequence of different roles that the 2 molecules play in regulation. RNAP is a molecule that requires flexibility in its binding, because it regulates the expression of >70% of all *E. coli* promoters (Tierrafría *et al.* 2022). Our results suggest that this functional requirement of RNAP is aided by the structure of its EM. For CI, which is supposed to bind only a few specific promoters and whose promiscuous binding elsewhere might even be deleterious, the EM is more constrained than predicted.

Put together, the constraints acting on gene expression phenotypes in our system arise from varied aspects of the system, such as the binding of TFs/RNAP to DNA and the free concentrations of these molecules in the cell. The factors that come from the MAK part of the model—concentrations of molecules—are responsible for setting the limit to the phenotypes that can be achieved (the envelope), while also affecting the manner in which mutations explore the phenotypic space (phenotypic volume). The factors that influence the TD part of the model—promoter architecture and the structure of the EMs—primarily affect how freely mutations explore that envelope. In other words, only the MAK predominantly constrain the maximum values of phenotypes achievable through mutations, while both parts of the model shape how easily those values are reached. This suggests that, if all we are interested in is the maximum range of phenotypic values but not how easily those phenotypes can be realized by mutations, it would be sufficient to reduce EMs, which are complex, high-dimensional mathematical objects, to a small number of summary statistics. We will return to this idea later in the manuscript.

## De novo promoter evolution—phenotypes and fitness of random sequences

So far, we examined how the 6 dynamical phenotypes are affected by mutations in an existing, functional Lambda $P_R$ promoter, allowing us to understand the mechanistic constraints shaping the evolution of already functional promoters. To understand de novo promoter evolution, we need to describe the phenotypes and fitness associated with random sequences, and then examine what trajectories they might take as they evolve under selection for regulation. In doing so, our approach is inspired by previous works on bacterial (Berg *et al.* 2004; Mustonen and Lässig 2005; Aguilar-Rodríguez *et al.* 2018) and eukaryotic (Stone and Wray 2001; Wray *et al.* 2003; Chen and Rajewsky 2007) gene regulatory elements, with the added complexity of examining multiple gene expression phenotypes as opposed to focusing only on steady-state expression levels or the underlying sequence variation.

Extending our model (Fig. 2) to any random sequence relies on our previous work that allowed predicting RNAP binding to any random sequence, and the assumption that the binding of CI to random sequences (as opposed to mutants around the given wild-type) can be predicted from its EM. We also assume that any random sequence can, in principle, act as a repressible promoter, if it binds RNAP and a repressor. We evaluated phenotypes associated with random sequences, which play a key role in determining evolutionary outcomes but are rarely done (some examples in bacterial regulatory elements: Horwitz and Loeb 1986; Yona *et al.* 2018; Lagator *et al.* 2022). We examined how the phenotypic effects of random sequences were distributed, in order to understand how likely a random sequence is to bind RNAP and CI. To explore the phenotypic effects of random sequences, we sampled $2 \times 10^9$ random 80 bp long sequences. For random sequences, we evaluated binding of both, RNAP and CI, to all possible configurations in each 80 bp sequence, meaning that we did not constrain their binding to any specific binding sites like in previous sections. We assumed that if a predicted CI binding site overlapped or was downstream of the predicted RNAP binding site, the system was repressed. In contrast, if no CI binding sites were predicted or if they were found upstream of the RNAP binding site, no repression occurred. The cooperativity between 2 CI molecules would stabilize their binding only if the 2 binding positions were 16–18 bp apart, mimicking the architecture of the wild-type $P_R$ promoter.

We found that for all 6 phenotypes, functional sequences were very rare with most random sequences being nonfunctional (Fig. 5a), as previously observed for simple regulated promoters (Maerkl and Quake 2007; Kinney *et al.* 2010) and proteins (Maerkl and Quake 2009; Jacquier *et al.* 2013). This finding was in contrast to the previous observations that random sequences often contain constitutive promoters (Horwitz and Loeb 1986; Yona *et al.* 2018; Lagator *et al.* 2022), suggesting that it is the more constrained protein (CI in our case, see Fig. 4d) that imposes the limits on the rates of promoter evolution.

To describe the fitness of a given random sequence, we assumed a quadratic fitness landscape with the phenotypic values of the wild-type Lambda $P_R$ defining the optimum. The fitness of each sequence was calculated as the distance of each of the 6 phenotypes (Fig. 5a) from the optimum ($P_R$) value of that phenotype, with each phenotype carrying equal weight. In this way, we assigned a fitness value to each of the $2 \times 10^9$ random sequences in order to characterize the distribution of fitness values across the entire genotypic space. The assumption of a quadratic fitness landscape might not capture how selection acts on promoter sequences, as it is more accurate near the assumed optimal wild-type than further away

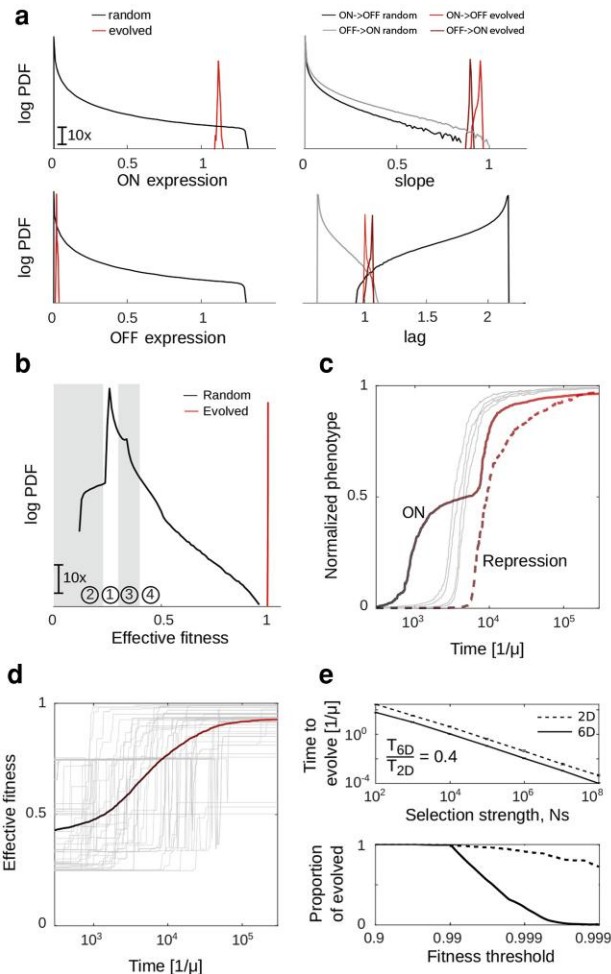

**Fig. 5.** Evolution of regulated promoters from random sequences. a) Phenotypic effects of random sequences, shown as the probability density function (PDF) for $2 \times 10^9$ fully random sequences (black and gray) or for 2,000 random sequences at the end of their simulated evolution toward $P_R$ promoter function (red and dark red). Distributions are shown for all 6 phenotypes: ON, OFF, ON->OFF lag, OFF->ON lag, ON->OFF slope, and OFF->ON slope. All phenotypic values are relative to Lambda $P_R$ phenotypic values, except the OFF expression, which is represented as the *repression ratio* (ratio of ON and OFF expression levels). b) Fitness distribution of $2 \times 10^9$ fully random sequences (black) or for 2,000 random sequences at the end of their simulated evolution toward $P_R$ promoter function (red). Each of the 6 phenotypes equally contributes to overall fitness, with the fitness peak defined as the Lambda $P_R$ values of each phenotype. The distribution can be split into 4 sections, defined with respect to the mechanistic functioning of the system: (1) ON = OFF = 0; (2) 0 < ON = OFF < $ON_{PR}$; (3) ON = OFF > $ON_{PR}$; (4) 0 < OFF < ON < $ON_{PR}$. c) Time trajectories of phenotypes, showing the order in which phenotypes evolve. Each curve represents a median of 2,000 sequences for which evolution was simulated. For a better comparison on the order of when phenotypes evolve, we normalized phenotypes to start at 0 and end at 1. See Supplementary Fig. 5 for nonnormalized results. The 4 gray lines are, in order left to right, ON->OFF lag, OFF->ON lag, OFF->ON slope, and ON-OFF slope. d) Simulated rates of promoter evolution for 2,000 random sequences, showing the large variance in the distribution of times taken for selection of Lambda $P_R$-like phenotypic values (time is shown in log units). The colored line is the average. e) Top: evolving promoters by selecting for only 2 phenotypes (ON and OFF states, referred to in the plot as "2D") is robustly almost 2-fold slower than selecting for all 6 phenotypes ("6D") across a wide range of selection strengths. Bottom: Proportion of evolved sequences as a function of fitness threshold (fitness value, relative to the wild-type Lambda $P_R$ phenotypic values, above which the promoter was considered evolved). With higher fitness threshold to consider a promoter evolved, increasing proportion of sequences get trapped in a local optimum, leading to lower proportion of evolved sequences. This effect is more pronounced when selecting all 6 phenotypes (6D).

from it and the phenotypes might not equally contribute to organismal fitness. In spite of the potential shortcomings, we rely on this model of fitness as it is commonly used in evolutionary biology and use it to illustrate how a mechanistic framework can be utilized to tackle biological complexity.

There is a general belief based on experimental findings (Sanjuan *et al.* 2004; Jacquier *et al.* 2013; Metzger *et al.* 2016; Duveau *et al.* 2017) that function among random sequences is vanishingly rare, although this view has been challenged (Yona *et al.* 2018; De Boer *et al.* 2020; Lagator *et al.* 2022). The mechanistic approach we adopted allowed us to quantify the probability of random sequences being functional, and we found that functional repressible promoters (those with phenotypes at least somewhat similar to the wild-type $P_R$ promoter) were rare, occurring with probabilities of $10^{-5}$– $10^{-7}$. While small, the likelihood of evolving a functional repressible promoter is orders of magnitude higher than it could be if TF and RNAP binding was less tolerant to sequence variations (as captured by their EMs).

## De novo promoter evolution—evolutionary trajectories

The distributions of phenotypes and fitness of random sequences describe the potential starting points for repressible promoter evolution. As a miniscule portion of random sequences acts as repressible promoters, selection must be involved for regulated promoters to emerge. To simulate such evolution, we started with 2,000 random, 80 bp long sequences and used an SSWM model adapted from Tuğrul *et al.* (2015). As mentioned above, optimal fitness was defined as the Lambda $P_R$ values of each of the 6 phenotypes.

We first asked whether the 6 phenotypes appeared in the simulated populations in a specific order. As expected, ON expression always emerged first, often orders of magnitude faster than other phenotypes. The emergence of RNAP binding before CI binding sites is not only predicated on the fact that repression cannot occur before expression, but also likely on the rapid emergence of RNAP binding sites in random sequences (Yona *et al.* 2018; Lagator *et al.* 2022). However, there was also an order in which other phenotypes emerged in the population (Fig. 5c; Supplementary Fig. 5), implying that selection for multiple phenotypes might be more predictable than expected, even though the time to evolve regulation is highly variable and unpredictable (Fig. 5d).

Intuitively, selecting for an additional phenotype might slow down evolution, because each phenotype needs to reach its own optimum. Starting from the same 2,000 random sequences, we compared how rapidly regulated promoter evolved when selection acted on all 6 phenotypes (6D) or only on 2 phenotypes (ON and OFF expression—2D). Interestingly, selecting on all 6 phenotypes led to more rapid rates of evolution, while being less precise (Fig. 5e): the populations selected in 6D more rapidly approached the optimum (Fig. 5e, top), but were less likely to reach the exact fitness of the wild-type $P_R$, compared to the populations selected only for ON and OFF expression (Fig. 5e, bottom), presumably because they were stuck on a relatively high but still suboptimal fitness peak.

The observed evolutionary dynamics stem from the constraints and correlations that characterize the GP landscape (Fig. 3). The constrained nature of the GP landscape of the evolving promoters means that a mutation that alters one phenotype is likely to alter other phenotypes as well. Early on during selection, most random sequences have very low fitness meaning that mutations are more likely to have a neutral or beneficial effect (Fisher 1930). Under such conditions, correlations between phenotypes mean that a mutation that is beneficial for 1 phenotype is also

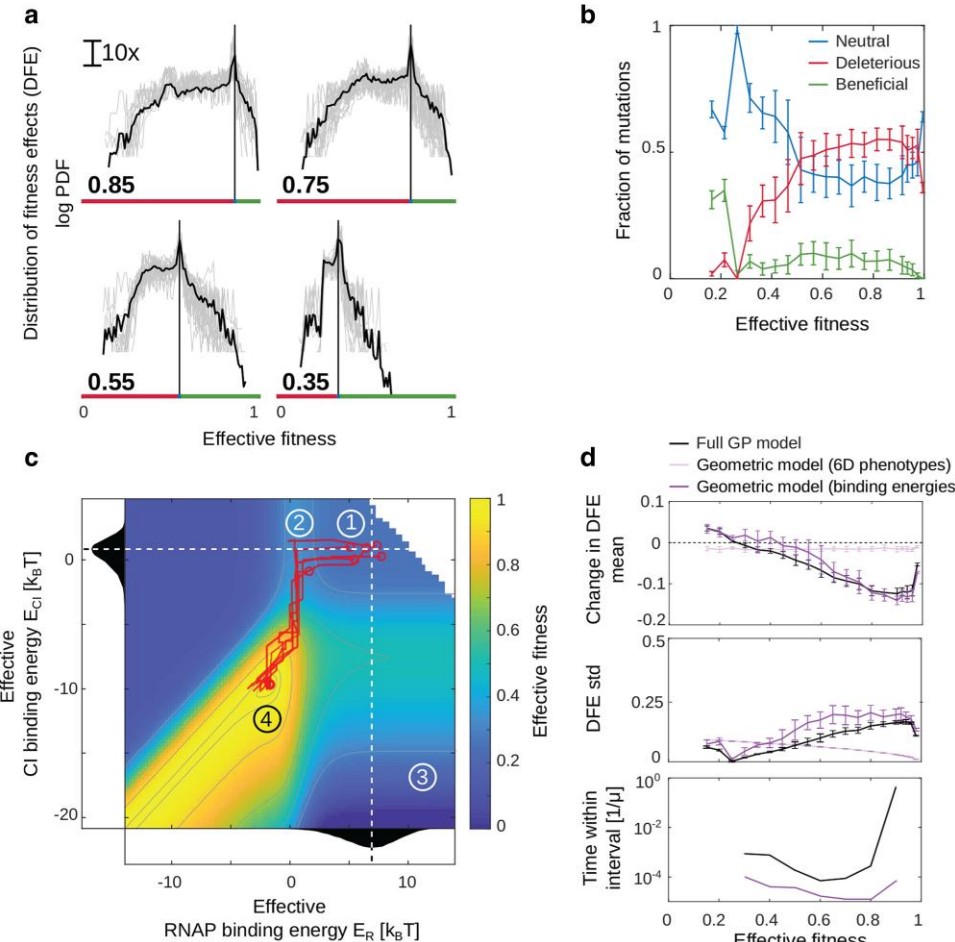

**Fig. 6.** Fitness effects of mutations as a function of promoter evolution. a) DFEs obtained by modeling the effect of every possible single and double point mutation for 100 replicate runs of evolving a promoter from a random sequence. DFEs were obtained at 4 different points along promoter evolution, when the evolving sequence had a fitness of 0.35, 0.55, 0.75, or 0.85 (indicated by the horizontal line) relative to the fitness of the Lambda $P_R$ promoter (defined to be the fitness optimum). Gray lines are all the DFE estimates of all 100 replicate evolving promoters, the black line is the mean. Red and green bars delineate beneficial (green) and deleterious (red) mutations. b) Proportion of neutral, beneficial, and deleterious mutations changes as the function of promoter fitness. Deleterious and beneficial mutations were defined as those mutations that alter fitness by at least 0.02 (which was the background noise value in our system). Error bars are standard error of the mean of 100 replicates. c) Representation of the fitness landscape along which random sequences evolve toward Lambda $P_R$ function. The landscape is represented with respect to the total binding energies between DNA and RNAP or CI (taking into account possible binding at multiple sites within the promoter). Contours show lines of equal fitness. Numbers 1–4 correspond to the mechanistically interpretable states of the evolving promoter, as seen in Fig. 5b. Red lines are individual example trajectory replicates for 6 evolving promoters, which all follow an ordering where the RNAP binding site evolves first, followed by the evolution of CI binding. Black distributions along the x- and y-axes are the binding energies of RNAP and CI to random sequences. d) Comparison of how the 3 models predict promoter evolution: (1) the full biophysical model, based on a comprehensive GP mapping (Fig. 2); (2) the traditional implementation of the Fisher's geometric model, where phenotypic values are drawn in 6D space at random from a fixed distribution without accounting for any underlying mechanisms; (3) a modified Fisher's geometric model, where we draw 2 biophysical parameters (binding energies of RNAP and CI)—instead of 6 phenotypes as in (2)—independently from a fixed distribution. Top left: the difference in mean of DFE and fitness of starting (random) genotype as a function of fitness. Negative values represent that the majority of mutations are deleterious. Top right: mean effect of mutations on ON expression as a function of fitness. Middle right: mean effect on OFF. Middle left: Standard deviation of DFEs as a function of fitness. Error bars represent standard deviation of mean and SD estimates over 100 replicates. Bottom: Time spent within the fitness interval of 0.1. The results are the median of 2,000 evolving sequences. Note that, while the values predicted by the geometric models depend on the specific parameters used, changes in these parameters would not affect the overall trends (for specific values we used, see Geometric model of Materials and methods).

more likely to be beneficial for 1 or more other phenotypes. However, as the population approaches the optimum, the likelihood of a mutation having a beneficial effect on any of the phenotypes decreases. Correlations between phenotypes when the population is near the optimum mean that when a mutation emerges that is beneficial for 1 phenotype, its overall effect on fitness might be diminished as it is more likely to have a negative effect on at least 1 other phenotype. In general, the closer the population is to the optimum, the more likely is the existence of correlations between phenotypes to slow down evolution, potentially explaining why the selection on 6 phenotypes was less likely to reach the precise fitness optimum.

## DFEs in evolving promoters

The manner in which a population can navigate a fitness landscape depends on the current genotype of that population. In other words, during the course of evolution as mutations are fixed in the population, the DFE can also change (MacLean *et al.* 2010; de Visser and Krug 2014; Seetharaman and Jain 2014; Couce *et al.* 2024). The mutational space of even a short sequence like a bacterial promoter is too large to explore comprehensively, posing a massive challenge for understanding how DFEs change during evolution. However, this problem would be significantly diminished if DFEs depended on the fitness of the organism and not just its genotype (Kryazhimskiy *et al.* 2009). In spite of the potential dependency of DFEs on current fitness, the

numerous experimental descriptions of DFEs almost exclusively focus on a given point in evolutionary time (Eyre-Walker and Keightley 2007; Soskine and Tawfik 2010; Kemble *et al.* 2019). We studied the relationship between current genotype, fitness, and DFEs by examining the fitness effects of all possible single and double mutations at several points during the simulated de novo promoter evolution of 2,000 random 80-bp sequences (see De novo promoter evolution—evolutionary trajectories).

The DFEs of evolving promoters changed during evolution (Fig. 6a). As the population moved toward its optimum, the frequency of deleterious mutations increased, although with a sharp decline for highly adapted promoters (Fig. 6b)—suggesting that the peak of the fitness landscape for repressible promoters contains a plateau. The width of the distribution also increased as the population approached the optimum, suggesting that mutations in promoters with higher fitness were more likely to drastically alter fitness, typically negatively (Fig. 6a).

Importantly, the DFEs of evolving promoters were largely dependent on the fitness of those promoters, rather than their specific genotype. The dependency of DFEs on fitness was strong and consistent across all 2,000 evolving sequences (Fig. 6a). Hence, the relationship between promoter DFEs and current fitness is determined by the mechanisms of promoter function in a manner that is largely independent of the starting random sequence or the particular mutations that get fixed. The consistency of promoter evolution is also captured by the observation that the proportion of beneficial mutations is relatively constant as the population evolves (except at very low fitness values) (Fig. 6b). Consequently, the structure of the fitness landscape and the manner in which populations navigate that landscape is surprisingly consistent and does not strongly depend on the particular starting sequence (Fig. 6c, red trajectories which all follow nearly the same path in the fitness landscape). This phenomenon, if generalizable to other systems (Kryazhimskiy *et al.* 2009; Aggeli *et al.* 2021), would result in more repeatable adaptation pathways that are defined predominantly by a single, scalar quantity (current fitness), which would dramatically reduce the complexity that must be accounted for to accurately describe evolution.

## DFEs in evolving promoters—functional models

Our model allowed us to investigate whether this consistency of promoter evolution observed across random sequences was caused by specific mechanisms of promoter function. To explore this question, we simulated the evolution of the same 2,000 random sequences toward the same fitness peak, but we determined the phenotypic effects of novel mutations during evolution in 3 different ways. The first represents "realistic" evolution and used the full model (Fig. 2), meaning that phenotypic effects of mutations were determined by the biophysical mechanisms underpinning promoter function. The second used a geometric model assigning each mutation a random effect on each of the 6 phenotypes in an independent and equally distributed manner, without accounting for how those mutations affect the binding energy first ("geometric model—6D phenotypes"). As such, this model is akin to methods typically employed in population genetics (Mustonen and Lässig 2005; Charlesworth and Charlesworth 2017), which assume the phenotypic (or fitness) effects of mutations without accounting for the underlying mechanisms that determine those effects. The third used a geometric model to select binding energies from an assumed distribution with the same range of values as the RNAP and Lambda CI EMs, but without accounting for the internal structure of the EMs ("geometric model—binding energies"). In other words, we assumed random energy penalties associated with every

mutation but then assigned values to the 6 phenotypes using the full model, meaning that the "geometric model—binding energies" accounted for the nonlinear relationship between binding energy and gene expression captured by the TD model, while the "geometric model—6D phenotypes" did not.

Not accounting for the relationship between the biophysics of protein-DNA binding and how it shapes the effects of mutations resulted in significantly different descriptions of promoter evolution (Fig. 6d; Supplementary Figs. 6 and 7). The underlying mechanisms introduce structure into the genotype-phenotype-fitness mapping that is critical for accurately capturing evolutionary dynamics. This is likely because the full model, compared to the "geometric model—6D phenotypes", determines phenotypic effects of mutations in a nonlinear fashion (Fig. 2), while also accounting for the constraints that emerge from the relationship between protein-DNA binding and the 6 promoter phenotypes (Fig. 3). However, the internal structure of the RNAP and Lambda CI EMs (i.e. how the energy penalties are actually distributed within the matrices) was not crucial for accurately modeling evolution, as the geometric model based on binding energies described similar evolutionary trajectories as the full model albeit with different time scale estimates. As such, the relative difference in the effects of individual mutations on the binding energy was not critical for modeling evolution, compared to accounting for the inherent mechanistic relationship between genotype and the gene expression phenotypes. In other words, to accurately model evolution (which we determined by comparing the 2 simplified models to the full model), it is (1) critical to account for the mechanism linking genotypic to phenotypic changes—in this case, the nonlinear relationship between binding energy and gene expression dictated by thermodynamics, and, therefore (2) sufficient to know only the range of values within the EMs, but not the actual distribution of values—dramatically reducing the number of variables needed to capture evolutionary trajectories from 152 (RNAP and CI EMs) to 2 (minimum and maximum energy values in those EMs). Therefore, the ability to simplify systems in this manner is contingent on knowing what level to draw the phenotypic effects from. Accurately capturing the evolution of bacterial promoters requires accounting for the effects of mutations on binding energies, rather than directly modeling evolutionary dynamics by assuming effects of mutations directly on gene expression phenotypes.

## Genotypes of the evolved promoters

From a theoretical perspective, predicting how a population traverses a given fitness landscape has received more attention than predicting the outcomes of evolution (Charlesworth and Charlesworth 2017; de Visser *et al.* 2018). This is, in large part, due to the lack of comprehensive GP maps, resulting in an advanced understanding of how selection operates but a relatively poor description of the genotypes that actually evolve.

The GP map of multidimensional gene expression phenotypes that we developed allowed us to not only understand how repressible promoters evolve but also what genotypes were favored by selection. Specifically, we were interested in what promoter architectures were more likely to emerge when random sequences evolved into repressible promoters. Informed by the architecture of most repressible bacterial promoters (Tierrafría *et al.* 2022), we considered CI binding sites that overlap with RNAP binding sites or are downstream of them to lead to repression. Then, we observed the number of CI binding sites that evolved (see Methods). We also observed the architecture of the promoters that emerged, defined as the relative position of the strongest (dominant) CI binding site relative to the RNAP binding site (Fig. 7a).

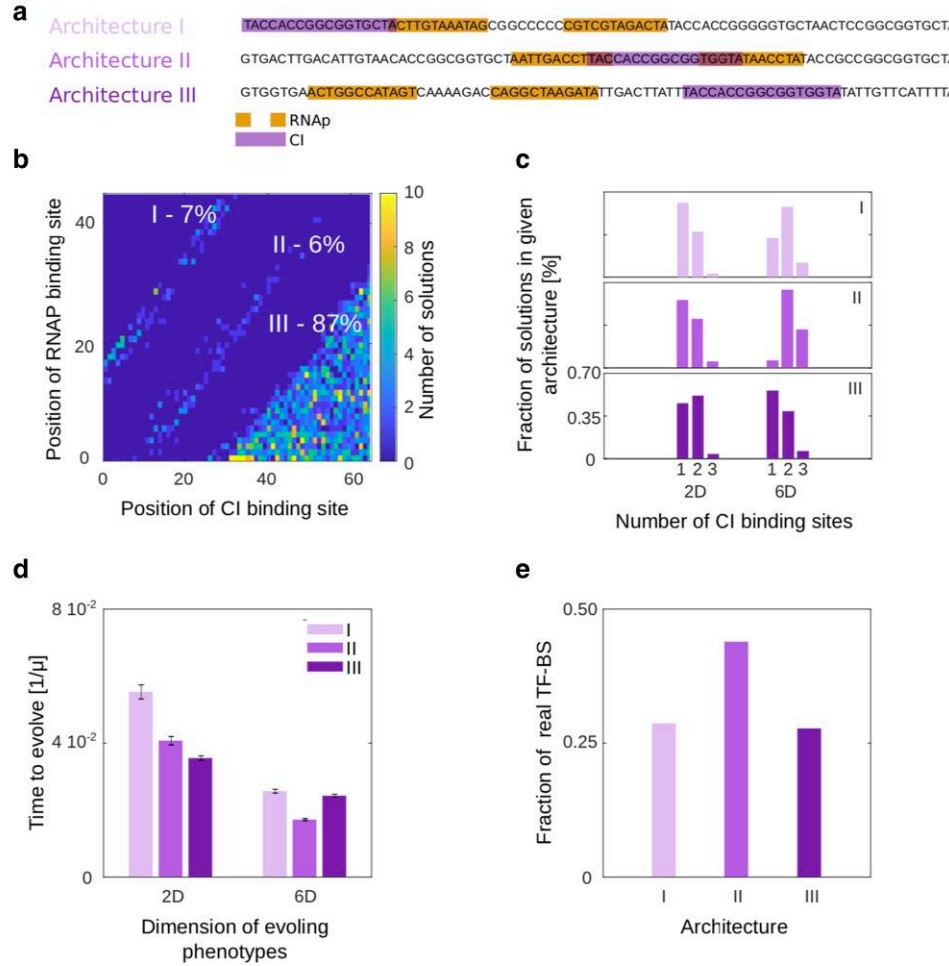

**Fig. 7.** Genotypic structure of evolved promoters. a) Example of the 3 possible promoter architectures that can emerge as random sequences evolve toward regulated promoters. These include the strongest CI binding site emerging upstream (architecture I), in between (architecture II), or downstream (architecture III) of the RNAP binding sites. b) We observed the distribution of promoter architectures following the evolution of 2,000 random sequences of 80 bp toward regulated promoters (i.e. toward Lambda $P_R$-like function). c) The number of CI binding sites that emerged in the 2,000 evolved promoters depended on the architecture that was evolving, as well as whether selection was acting on 2 (only ON and OFF) or all 6 phenotypes. d) Time to promoter evolution, measured in $1/\mu$ where $\mu$ is the mutation rate and based on the evolution of 2,000 sequences with population size $N = 10^6$. Error bars are standard deviations across all promoters that evolved that architecture. Time to promoter evolution depended on the architecture that was evolving, as well as whether selection acting on 2 or all 6 phenotypes. e) Using the data from RegulonDB, we obtained the position of binding relative to the RNAP binding site of all known *E. coli* repressible promoters (over 700). These were classified in one of the 3 architectures shown in panel a).

The likelihood of emergence was not random for the 3 promoter architectures—the dominant CI binding site was more likely to appear downstream of the RNAP binding site rather than overlapping with it (Fig. 7b). The evolved promoter architecture (i.e. the location of the strongest binding site) also impacted the total number of CI binding sites that needed to evolve in order to reach wild-type $P_R$ levels of repression (Fig. 7c). The likelihood of a given architecture emerging was related to its speed of evolution (Fig. 7d) which was, at least in part, affected by the constraints associated with that architecture (Fig. 4c). Furthermore, the RNAP binding site, which always evolves first (Fig. 5c), introduced further constrains on the emergence of CI binding site(s). For example, when the dominant CI binding site evolved between the RNAP −10 and −35 binding site (architecture II), it often required additional CI binding sites to reach the fitness optimum (Fig. 7c). This is because the direct overlap between CI and RNAP binding sites limited the range of mutations that could increase CI binding (i.e. increase fitness) without negatively affecting RNAP binding (which would decrease fitness).

Selection acting only on 2 phenotypes (ON and OFF) predicted different evolutionary outcomes (promoter architectures and binding site numbers) to selection acting on all 6 phenotypes, with 6D selection resulting more frequently in multiple CI binding sites (Fig. 7c). Furthermore, the predicted rates of evolution of the 3 architectures were also different between selection for 2 vs all 6 phenotypes (Fig. 7d).

It remains unexplored whether selection in repressible promoters actually acts on dynamical (6D) or only on steady-state (2D) phenotypes. To indirectly examine this question, we collected the information about all known promoters in *E. coli* from RegulonDB (Tierrafría *et al.* 2022). Specifically, we classified all known repressible promoters into the 3 promoter architectures (Fig. 7a), using the information about the known position of repressor binding sites relative to the RNAP binding sites. Interestingly, we found that the largest number of known promoters had a repressor binding site between the −10 and −35 RNAP sites (architecture II) (Fig. 7e). Our model predicted this architecture to arise most rapidly when selection acts on 6 phenotypes, but not when it acts on only 2 (Fig. 7d). Therefore, while a multitude of factors likely contributed to

## Discussion

The main aim of this study was to serve as a primer on how mechanistic models can be utilized to account for and tackle molecular and biophysical complexity in the context of evolution. Even in simple biological systems like ours, consisting of 3 molecular components (RNAP, CI, and the promoter), the mapping of genotype onto phenotypes onto fitness is extremely complex due to the mutational space being prohibitively large; a change in genotype potentially altering multiple cellular phenotypes; the unclear and gene-specific relationship between phenotypic change and fitness; and the dependency of phenotype-fitness mapping on the environment. A natural consequence of this complexity is the drive to focus on studying *specific* examples—the forms that survived competition and selection—in order to understand the strategies that made them successful. This comes at the cost of not understanding *why* any of the myriad other possible forms did not arise or get fixed in a population. To paraphrase François Jacob: evolutionary biology is focused on studying the "actual" at the cost of not understanding the "possible" (Jacob 1977). In this work, we build on previous works and ideas (Ackers *et al.* 1982; Dean and Thornton 2007; Vilar 2010; Josephides and Moses 2011; Pai *et al.* 2015) to put forward a primer to resist this drive, by utilizing a mechanistic model to remain tied to the actual while enabling the exploration of the possible.

Besides our main aim of providing a primer for the analysis of complexity in biology and evolution from a mechanistic starting point, we also provided a range of novel findings about the evolution of bacterial promoters. We identified the mechanistic constraints acting on GP mapping in promoters, with the envelope (total range of possible phenotypes) set by the MAK and its coverage (manner in which that space is explored through point mutations) by the EMs of the DNA binding proteins (Fig. 3). Various cellular and genetic parameters, such as the concentration of the repressor or the architecture of the promoter, also affected the nature of GP mapping. Selection for constitutive expression evolves fast (Yona *et al.* 2018; Lagator *et al.* 2022) while regulation takes longer (Fig. 5), although if regulation is selected for based on the 6 gene expression phenotypes it might proceed faster than if selection acts only on steady-state expression levels (Fig. 7d). The outcomes of evolution, in terms of the promoter architectures that evolved, also depended on whether selection was acting on 6 or 2 phenotypes (Fig. 7c), These, and various other findings we reported, were made possible by our use of a mechanistic model to study promoter evolution, as it allowed dissecting the contribution of individual factors in a manner that would be difficult to replicate experimentally or with a more generic model.

The observation that quantitative descriptions of at least some key evolutionary properties require only a handful of parameters gives credence to the idea that extending the findings based on the modeling of Lambda $P_R$ promoter to other regulated promoters, or even to more complex networks, might be relatively straightforward (Vilar 2010; Josephides and Moses 2011). Modeling-wise, the fundamental aspects of the MAK part of the model ought to be true for any dynamical molecular system (Chen *et al.* 2010). Similarly, utilizing thermodynamics to construct a model for any promoter or a network of any size is possible (Bintu, Buchler, Garcia, Gerland, Hwa, Kondev, Kuhlman, *et al.* 2005; Bintu, Buchler, Garcia, Gerland, Hwa, Kondev, and Phillips 2005). The major difficulty in accurately mapping

genotype to phenotype in other networks comes from the fact that the predictive power of the TD model relies on having the relevant EMs (Kinney *et al.* 2010; Vilar 2010), and obtaining EMs is labor- and time-intensive (Barnes *et al.* 2019; Ireland *et al.* 2020). And yet, our results suggest that for understanding many aspects of promoter evolution, using easy-to-derive summary statistics of EMs (their extremal values) might be sufficient, providing a key insight into how our model can be extended to other systems. This is why not only describing GP mapping, but also understanding its mechanistic origins, ought to form a crucial and major direction in studying evolution as it allows parsing which components of a complex biological system are the key drivers of its evolution.

However, relying on a mechanistic model also meant that, somewhat ironically, we had to make a lot of concessions and simplifications that limit the generalizability of our conclusions. Gene expression levels, both at the transcriptional and especially at translational levels, are not determined only by the binding energies between TFs and DNA. In fact, numerous other mechanisms involved in the regulation of gene and protein expression levels have been described, and accounting for them forms the obvious extension of the work presented here. Furthermore, both components of our model—TD and MAK—are deterministic and do not account for the temporal and between-individual stochasticity in gene expression, which can play a role in function and evolution of regulatory systems (Arkin *et al.* 1998; Rosenfeld *et al.* 2005; St-Pierre and Endy 2008; Metzger *et al.* 2015). The model is also not calibrated to account for the effects of random mutations as accurately as the effects of the mutations around a given (wild-type) promoter sequence (Vilar 2010). Incorporating additional mechanisms involved in gene regulation can extend the mutational space in which the model predictions are accurate (Lagator *et al.* 2022), with more studies in this direction needed.

While we demonstrate the importance of accounting for how DFEs change as a function of fitness (Fig. 6a), we also use DFEs as a simplified description of the fitness landscape as a whole. In reality, fitness landscapes are much more complicated. One important contributor to fitness landscape structure is the interactions between mutations (Phillips 2008; de Visser and Krug 2014), and our model accounts for just 1 source of such epistatic interactions (Lagator, Paixão, *et al.* 2017). In particular, our model does not account for sign epistasis, which can occur between mutations in a promoter (Lagator *et al.* 2016) and is a major contributor to fitness landscape complexity (Poelwijk *et al.* 2011). Similarly, our work does not account for any potential interactions between mutations in promoters and TFs, which can have unpredictable consequences (Lagator, Sarikas, *et al.* 2017). Furthermore, not each phenotype is going to equally contribute to fitness like we assumed in this work. In other words, some of our 6 phenotypes might contribute more to fitness than others. The complex interaction between phenotype and fitness will always be dependent on the specific properties of the molecular system and the organism that are under selection imposed by particular environmental conditions.

We also assumed that the fitness landscape is quadratic. We have no experimental access to the actual, environment-dependent fitness function of our system. Moreover, our system is a synthetically simplified version of the natural lambda switch. Hence, we needed to make an assumption for fitness in lieu of actual experimental measurements and selected a commonly used, quadratic model that stabilizes phenotypes at their WT values corresponding to a single peak in the phenotype space. This model has 4 attractive features: (1) phenotypes with highest fitness correspond, by construction, to a functional regulated promoter (high ON state, low OFF state, WT dynamics); (2) this model is a theoretically

well-understood approximation for generic single-peaked fitness functions close to their fitness optimum (although we also simulate far away from the peak); (3) the same mathematical form of the fitness function generalizes to phenotypes of different dimensions, allowing us to compare how the dimensionality of the phenotype affects our conclusions (Fig. 5); and (4) this choice of the fitness function naturally connects to the geometric model (Fig. 6). Quadratic fitness is a minimal model with these properties, in the sense that any nontrivial results on evolutionary dynamics (e.g. possible multiple fitness peaks in the genotype space and sign epistasis) must be a consequence solely of the genotype-to-phenotype map rather than of the fitness function defined on top of any phenotypes. In other words, while modeling more complicated fitness functions would certainly be possible and possibly more realistic, such models would introduce additional structure(s) for which we have no justification in experiment or published literature.

Tackling complexity in evolution includes 2 key challenges. First is generating or simulating data that captures that complexity. Our approach was to develop a mechanistic model of gene expression dynamics from a repressible bacterial promoter that accounted for well-understood physical and chemical properties of proteins and DNA (while, inevitably, making various simplifications). The model allowed us to obtain estimates of phenotypes and fitness from a large number of random (neutral) and selected genotypes, resulting in a more realistic genotype-phenotype-fitness mapping than commonly used when modeling evolution, with potentially important consequences for the understanding of basic evolutionary properties (Hledík et al. 2022).

Importantly, the mechanistic model enabled addressing the second challenge in tackling complexity in evolution: how can the complexity be reduced without sacrificing evolutionary detail. To explore the ways in which mechanistic models can be utilized to address this challenge, we assumed that the full model could capture promoter evolution. That assumption, which we openly acknowledge is likely incorrect due to the various shortcomings of our model as discussed above, nevertheless provided us with a baseline for comparison of evolutionary predictions from various simplified (in terms of the number of parameters) versions of the model. We found that, in our system, accurate description of several key evolutionary properties was possible without accounting for every parameter that contributed to them. For example, understanding what phenotypes can be accessed through mutations does not require examining every possible mutant. In fact, the total range of phenotypic values accessible through mutations (*envelope* in Fig. 3) can be discerned by knowing only the minimum and maximum effects a mutation could have, as it did not require accounting for the discreteness of individual mutation effects. Similarly, only the total range of mutational effects (minimum and maximum EM values) on the phenotype was required to accurately capture the evolutionary dynamics of promoters (Fig. 6d). In other words, the biophysically realistic link between mutations and their phenotypes, even a simplified one, was sufficient to alter predictions of evolutionary dynamics compared to assigning fitness effects directly from genotype as typically done when modeling evolution (Mustonen et al. 2008; Charlesworth and Charlesworth 2017). The reduction in complexity needed to capture the evolutionary dynamics was also justified when considering the fitness effects of mutations during the course of evolution, as the distributions of fitness effects largely depended only on the fitness of the current genotype (Fig. 6b). In contrast to most models of evolutionary dynamics, the mechanistic approach we used is not generic but rather refers to a specific system, meaning that the above predictions can be empirically tested.

Our aim was to develop a primer on how to tackle molecular complexity in order to better understand evolution. Relying on a mechanistic model provided novel insights into the regulation of regulated promoters. More importantly, utilizing a mechanistic model allowed us to understand when the complexity of the studied system could be reduced without sacrificing the understanding of how that system evolves. As such, we show how building effective, mechanistic models that capture aspects of the underlying molecular complexity can provide critical insights into what intermediate phenotypes, nonlinear interactions and constraints must be considered, and which can be ignored, if we want to get an accurate picture of how a biological system evolves.

## Data availability

Strains and plasmids are available upon request. All experimental raw data can be found in Supplementary Data 1. All code underpinning this work is available at https://github.com/Lagator-Group/Linking-Molecular-Mechanisms-to-their-Evolutionary-consequence-a-primer or upon request.

Supplemental material available at GENETICS online.

## Acknowledgments

The authors thank Nick Barton, Stepan Denisov, Claudia Igler, Srdjan Sarikas, Anna Staron, and the anonymous reviewers for useful comments and discussions that helped improve our work.

## Funding

Funding for this work was provided by the Wellcome Trust–Royal Society Sir Henry Dale Fellowship (216779/Z/19/Z) and the Royal Society Research Grant (RG\R2\232522) to M.L.

## Conflicts of interest

The author(s) declare no conflict of interest.

## Author contributions

Conceptualization and Writing—review & editing: R.G., C.C.G., G.T., and M.L. Methodology, Investigation, and Visualization: R.G., G.T., and M.L. Funding acquisition and Supervision: C.C.G., G.T., and M.L. Writing—original draft: R.G. and M.L.

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

*Editor: A. Moses*