## [Peer Review File · Genetics]

Linking Molecular Mechanisms to their Evolutionary Consequences: a primer

Rok Grah, Calin Guet, Gasper Tkačik, and Mato Lagator

NOTE: The reviews and decision letters are unedited and appear as submitted by the reviewers.

In extremely rare instances and as determined by a Senior Editor or the EIC, portions of a review may be redacted. If a review is signed, the reviewer has agreed to no longer remain anonymous.

The review history appears in chronological order.

Review Timeline:

Submission Date:	2024-04-09
Editorial Decision:	2024-07-09
Resubmission Received:	2024-09-24
Accepted:	2024-11-13

July 9, 2024

GENETICS-2024-307009

Linking Molecular Mechanisms to their Evolutionary Consequences: a primer

Dear Dr. Lagator:

Thanks for your patience while we have been handling this manuscript. In the end, we have still not heard back from two of our reviewers, despite multiple reminders and deadline extensions. Rather than searching for new reviewers at this stage, at the request of the Editor in Chief, I went ahead and reviewed the paper.

While your manuscript is not currently acceptable for publication in GENETICS, we would welcome a substantially revised manuscript. Both I and the reviewer are very supportive of the manuscript, and although there are several revisions that I feel are essential, I believe there should be no problem for you to address them. You can see my comments below.

We look forward to receiving your revised manuscript. Please let the editorial office know approximately how long you expect to need for revisions.

Upon resubmission, please include:

1. A clean version of your manuscript;
2. A marked version of your manuscript in which you highlight significant revisions carried out in response to the major points raised by the editor/reviewers (track changes is acceptable if preferred);
3. A detailed response to the editor's/reviewers' feedback and to the concerns listed above. Please reference line numbers in this response to aid the editor and reviewers.

Your paper will likely be sent back out for review.

Additionally, please ensure that your resubmission is formatted for GENETICS

<https://academic.oup.com/genetics/pages/general-instructions>

Follow this link to submit the revised manuscript: Link Not Available

Sincerely,

Alan Moses
Associate Editor
GENETICS

Approved by:
Audrey Gasch
Senior Editor
GENETICS

Reviewer #2 (Comments for the Authors (Required)):

This manuscript by Grah et al. presents a comprehensive study of genotype-phenotype-fitness relationships, using a combination of biophysical modeling, evolutionary simulations and experimentation. The modeling, combining thermodynamics-based and mass action kinetics models, is rigorous. The systematic characterization of constraints on phenotypic exploration through mutations is innovative and enlightening. Elucidation of mechanisms underlying those constraints, e.g., how promoter architecture and site overlap influence them, offers very interesting and sometimes unexpected insights. Simulations are used to study the evolution of gene regulation within this highly well-defined mechanistic framework, revealing very interesting findings about how individual phenotypes evolve when the fitness depends not only on those phenotypes but others correlated with them. The last section examining the nature of promoter architectures more likely to evolve under different assumptions is also novel. The paper is very well written, and the high-level interpretations and contextualization provided for the different parts of the analysis are spot-on and a joy to read. I only have some minor comments and suggestions for improvement of the manuscript, as noted below.

Specific comments:

Line 263: "When we fitted, rather than predicted, steady-state expression levels, the model fit dramatically improved for all mutants (Fig.2C,D).": This statement is not entirely clear. Does this mean that the TD parameters were trained on WT + 9 mutants and the MAK parameters were those learnt from the WT data alone?

Lines 398-401: "while not affecting the manner in which mutations explore the envelope (Fig.4A,B)". This is not clear. Figure 4B seems to show that changing the parameters in question does in fact change the "extent to which mutations evenly explore the space within the envelope". Clarification of this point is important as it also relates to the overall summary of how MAK parameters and TD parameters differ in their influence, as articulated in the last paragraph of that subsection.

The results presented in Figure 3 nicely address the question of constraints acting on genotype-phenotype mapping. One thing that is not as clear is what to make of the quantification being done in 2D spaces formed by pairs of phenotypes. The quantities reported here will be larger if constraints were examined on each phenotype separately, or smaller if the combination of 3 or more phenotypes was examined simultaneously. If this is true, stating it explicitly may help the reader take home the intended message from this subsection.

Figure 6D compares the evolutionary dynamics under the full mechanistic model to those under geometric models. Don't the inferences drawn from this depend on the parameterization of the geometric model?

Line 1242: "haven't" -> "have not". Line 1307: "doesn't" -> "does not". Line 519: "contract" -> "contrast". Line 530: "it's" -> "it is". Line 1347: "it's" -> "its". Line 1349, 1350: "didn't" -> "did not". (Please correct other such typos, not enumerating all here.) Please check line 756, the word "addition" seems out of place.

Associate Editor Comments:

This study develops a mechanistic model for a well-studied promoter, and investigates several interesting questions about how "energy models" from high-throughput data can be used to help understand the evolution of regulatory sequences and whether the "genotype-phenotype" map can be simplified to yield generalizable predictions.

Overall, I found the paper complicated and difficult to read. The introduction contains many general statements that while interesting, are not really supported by evidence. Thus, the introduction reads to me as a mixture of an opinion paper and a research paper. I encourage the authors to move the opinion portions to the Discussion (or simply remove them). Personally, I agree with many of the authors' opinions and believe they are taking steps in the right direction. (Indeed, we took a similar approach in my group many years ago, PMID: 21294912). However, the introduction as written tends to give a false impression of the progress made in this study. For example, in the introduction the authors' discuss the general problem of lack of generalizability of models. At the end of the paper, we finally get a test of one general prediction, and it's very exciting that they find some evidence for it. But in the end, it's only one prediction. So there is still a long way to go. Hence, I suggest the comments about generalizability of evolutionary models should be moved to the discussion as "future directions" for this type of research. Other examples can be found in my specific comments below.

Also, there is an overall lack of reference to relevant research, especially relating to modelling of bacterial promoters/gene expression and evolution of regulatory sequences. Again, specific examples are given below.

Major comments.

1. "We found that several key evolutionary³¹ properties - the distribution of phenotypic and fitness effects of mutations, the³² evolutionary trajectories during selection for regulation - can be accurately captured³³ without accounting for all, or even most, parameters of the system"

This is one of the most interesting results/claims of the paper, but I have two comments. First, the authors should discuss is to what extent these results depend on the assumption of a quadratic fitness landscape.

Second, I found it extremely difficult to understand what exactly was the evidence that supports this claim. I believe that the analysis is reported in Figure 6, and the authors write:

"to accurately model evolution it is sufficient to know only⁷⁰⁴ the range of values within the EMs, but not the actual distribution of values"

However, immediately after this, they write:

"The ability to simplify systems in this manner is contingent on drawing⁷⁰⁸ the effects of mutations from the correct distribution"

which seems to contradict the previous statement. How can we draw the effects of mutations from the correct distribution without knowing the actual distribution? Perhaps these are two different distributions? They conclude:

"Accurately capturing the evolution of bacterial promoters requires accounting for the effects of mutations on binding energies, rather than directly assuming their phenotypes"

I cannot figure out what this means. It doesn't seem consistent with the claim about the simplification of the model (earlier in this section). I strongly encourage the authors to connect their conclusions specifically to what they actually found (e.g., "Accurately capturing the evolution": as far as I can tell, no measurements of accuracy are reported, capture of evolution is a weak analogy at best.)

Figure 6 shows the results of complicated experiments, but it is not clearly stated how much of the variance in the DFEs/evolutionary trajectories is explained by the simpler model (152 vs. 2 parameters)? This effect should be quantified.

2. Overly general comments/lack of citations

-The authors study evolution of promoters from random sequences, but omit citations to studies of evolution of regulatory sequences from random. Going back at least to Stone & Wray 2001 (PMID: 11504856) and many subsequent papers. Clearly the work here is an advance, but the previous work should be discussed.

"On the one hand, molecular causes of evolutionary adaptations are frequently characterized, with numerous studies identifying the specific mutations that underpin a given phenotypic or fitness change observed in nature or in the laboratory. On the other hand, evolutionary biologists focus on the global properties of complex systems and their consequences for evolution, especially at the population and species levels."

I think I get the point, but we need to have specific examples of studies of both types. I also think these comments can be moved to the discussion, as they aren't really needed to introduce the current work.

"A similar split can be observed when considering models of evolution, which tend to either be abstract and general but lacking a direct connection to underlying molecular and cellular mechanisms; or are built to describe a specific system, but do not readily generalize beyond it and the experimental measurements they were designed to explain"

I think these comments are too general. We need specific citations of the types of models of evolution that the authors have in mind. For example, readers of GENETICS will likely know about continuous time Markov models of molecular evolution used for DNA and proteins that have no difficulty generalizing to very large numbers of sequences.

Also, the authors do great work to parameterize their model based on experimental measurements. However, in what sense can their model generalize beyond these measurements? In the end, there's only one qualitative prediction made. I think these comments should be moved to the discussion and can be considered as "future directions".

"In addition, following environmental change that affects the concentration of the TF in the cell, regulated promoters go through a transient phase before reaching a new steady-state expression level (Longo and Hasty 2006; Yosef and Regev 2011)"

The authors should add current references, especially this one seems relevant here: Cis-regulatory variants affect gene expression dynamics in yeast. PMID: 34369376

"The main aim of this work is to provide a primer on how to study biological systems of increasing complexity in order to understand their evolution"

I don't think "primer" is the right word. The authors use it many times in the paper, but after reading the paper, I am not sure what I have been "primed" for.

"that function among random sequences is vanishingly rare. "

De Boer's papers challenge this (at least for yeast). Worth citing here.

"utilizing a mechanistic model to remain tied to the actual while enabling exploration of the possible"

My group did something similar in our paper: "Modeling the evolution of a classic genetic switch" PMID: 21294912

"extending the findings based on the modelling of Lambda PR promoter to other854 regulated promoters"

Certainly, this approach can be applied to other promoters. In our paper (PMID: 21294912) we also tried to predict the genotype-phenotype relationship in a simple regulatory system using a Shea & Ackers framework, summarized the phenotype using a few dimensions and then studied evolution of the network.

I think our work should be discussed.

Minor essential comments

1. "The mass action kinetics (MAK) uses standard ODEs to describe the temporal240 dynamics of molecular concentrations in a system. In our system, MAK accounts for241 the changes in concentrations of the repressor CI and the measurable system output,242 YFP. "

The authors should at the very least discuss the influential literature relating to stochasticity in this promoter (and gene expression more generally). This goes back to at least Arkin & McAdams 1998 (PMID: 9691025), but has been considered many times since (PMID: 15790856, PMID: 19098103, PMID: 21119634). Some arguments about why they believe they can study evolution of this system using a deterministic approximation would be beneficial here.

2. "When we fitted, rather than predicted, steady-state263 expression levels, the model fit dramatically improved for all mutants (Fig.2C,D). Our264 model therefore accurately predicted the effects of Lambda PR promoter mutations on265 the dynamics of gene expression, with the error largely stemming from the266 comparatively poorer predictions of steady-state expression levels"

This argument is not correct from a statistical perspective unless the authors know that they are fitting the "true" model and the parameter estimates are independent. Since many features of gene expression have been approximated and the model is complex, these assumptions are unlikely to be true. (Consider fitting a line to a polynomial. Surely you can improve the fit of the line by adjusting the intercept. But this does not imply that the error results only from an incorrect guess of this parameter.)

3. "In other words, some changes to gene expression phenotypes cannot be340 achieved by mutating the promoter and might have to rely on changes to other341 components of gene regulation machinery (such as TFs)"

As far as I understand it, the authors only considered single and double mutations in the promoter. To make this claim, they would need to rule out indels, duplications, horizontal gene transfer, etc. The claim is unnecessary and can just be cut.

4. "This finding456 suggests that the internal structure of the EMs might have been selected for."

"Put together, the constraints acting on genotype-phenotype mapping arise from varied465 aspects of the system"

"o describe the fitness of a given random sequence, we assumed a quadratic fitness534 landscape (a common model of fitness effects in evolutionary biology), with the535 phenotypic values of the wild type Lambda PR defining the optimum."

These comments seem too much. The authors are considering only a single promoter in isolation. We have no idea to what extent this reflects the biological function of this promoter in the context of the intact regulatory network. In other words, the authors are considering gene expression dynamics of a single promoter as a "phenotype" but have not actually connected it to fitness. The assumption of a quadratic fitness function seems ok for phenotypes very near the wt, but is probably not very good for phenotypes far from the wt.

Dear Dr. Lagator:

Thanks for your patience while we have been handling this manuscript. In the end, we have still not heard back from two of our reviewers, despite multiple reminders and deadline extensions. Rather than searching for new reviewers at this stage, at the request of the Editor in Chief, I went ahead and reviewed the paper.

While your manuscript is not currently acceptable for publication in GENETICS, we would welcome a substantially revised manuscript. Both I and the reviewer are very supportive of the manuscript, and although there are several revisions that I feel are essential, I believe there should be no problem for you to address them. You can see my comments below.

We look forward to receiving your revised manuscript. Please let the editorial office know approximately how long you expect to need for revisions.

Upon resubmission, please include:

1. A clean version of your manuscript;
2. A marked version of your manuscript in which you highlight significant revisions carried out in response to the major points raised by the editor/reviewers (track changes is acceptable if preferred);
3. A detailed response to the editor's/reviewers' feedback and to the concerns listed above. Please reference line numbers in this response to aid the editor and reviewers.

Your paper will likely be sent back out for review.

Additionally, please ensure that your resubmission is formatted for GENETICS <https://academic.oup.com/genetics/pages/general-instructions> [academic.oup.com]

Follow this link to submit the revised manuscript: <https://genetics.msubmit.net/cgi-bin/main.plex?el=A7NR1GKn2A6aYN4I2A9ftdZPgJdZnrYeqTJe6Tb9jgZ> [genetics.msubmit.net]

Sincerely,

Alan Moses
Associate Editor
GENETICS

Approved by:
Audrey Gasch
Senior Editor
GENETICS

We thank the Editors and the Reviewer for your support of our work. We are including a substantially revised manuscript in which we addressed all the comments raised. Point-by-point responses to comments are below.

Reviewer #2 (Comments for the Authors (Required)):

This manuscript by Grah et al. presents a comprehensive study of genotype-phenotype-fitness relationships, using a combination of biophysical modeling, evolutionary simulations and experimentation. The modeling, combining thermodynamics-based and mass action kinetics models, is rigorous. The systematic characterization of constraints on phenotypic exploration through mutations is innovative and enlightening. Elucidation of mechanisms underlying those constraints, e.g., how promoter architecture and site overlap influence them, offers very interesting and sometimes unexpected insights. Simulations are used to study the evolution of gene regulation within this highly well-defined mechanistic framework, revealing very interesting findings about how individual phenotypes evolve when the fitness depends not only on those phenotypes but others correlated with them. The last section examining the nature of promoter architectures more likely to evolve under different assumptions is also novel. The paper is very well written, and the high-level interpretations and contextualization provided for the different parts of the analysis are spot-on and a joy to read. I only have some minor comments and suggestions for improvement of the manuscript, as noted below.

We thank the Reviewer for the positive appraisal of our work and the kind words.

Specific comments:

R2.1 Line 263: "When we fitted, rather than predicted, steady-state expression levels, the model fit dramatically improved for all mutants (Fig.2C,D)": This statement is not entirely clear. Does this mean that the TD parameters were trained on WT + 9 mutants and the MAK parameters were those learnt from the WT data alone?

We changed the text to clarify that all the model parameters were obtained only from the WT, not the mutants (L304 – in document with tracked changes).

R2.2 Lines 398-401: "while not affecting the manner in which mutations explore the envelope (Fig.4A,B)". This is not clear. Figure 4B seems to show that changing the parameters in question does in fact change the "extent to which mutations evenly explore the space within the envelope". Clarification of this point is important as it also relates to the overall summary of how MAK parameters and TD parameters differ in their influence, as articulated in the last paragraph of that subsection.

We thank the Reviewer for identifying this unclear statement. What we meant was that one could change the envelope (for example, stretch it to be 3x bigger) but the distribution of the mutations across the different categories shown in Fig.3 would not change. We recognize this was unclear and have amended the text to clarify (473-476).

R2.3 The results presented in Figure 3 nicely address the question of constraints acting on genotype-phenotype mapping. One thing that is not as clear is what to make of the quantification being done in 2D spaces formed by pairs of phenotypes. The quantities reported here will be larger if constraints were examined on each phenotype separately, or smaller if the combination of 3 or more phenotypes was examined simultaneously. If this is true, stating it explicitly may help the reader take home the intended message from this subsection.

We agree with the Reviewer and have added an explanation in the figure legend (L448-451).

R2.4 Figure 6D compares the evolutionary dynamics under the full mechanistic model to those under geometric models. Don't the inferences drawn from this depend on the parameterization of the geometric model?

The Reviewer is correct that the values estimated would depend on the specific parametrization. However, the key finding we present is that the predictions from the geometric model with phenotypes are drastically different to the 'full model', compared to the relatively modest differences in the predictions arising from the geometric model on binding energies. As such, both use the same parameters of the geometric model. The differences in their predictions arise from the geometric model on binding energies including non-linearity of the thermodynamic model when

assigning phenotypes, while applying the geometric model directly on phenotypes does not do this. We added additional explanation to clarify this point (L857-859).

R2.5 Line 1242: "haven't" -> "have not". Line 1307: "doesn't" -> "does not". Line 519: "contract" -> "contrast". Line 530: "it's" -> "it is". Line 1347: "it's" -> "its". Line 1349, 1350: "didn't" -> "did not". (Please correct other such typos, not enumerating all here.) Please check line 756, the word "addition" seems out of place.

We thank the Reviewer for pointing to these typos, we have changed all of them and identified additional ones.

Associate Editor Comments:

This study develops a mechanistic model for a well-studied promoter, and investigates several interesting questions about how "energy models" from high-throughput data can be used to help understand the evolution of regulatory sequences and whether the "genotype-phenotype" map can be simplified to yield generalizable predictions.

Overall, I found the paper complicated and difficult to read. The introduction contains many general statements that while interesting, are not really supported by evidence. Thus, the introduction reads to me as a mixture of an opinion paper and a research paper. I encourage the authors to move the opinion portions to the Discussion (or simply remove them). Personally, I agree with many of the authors' opinions and believe they are taking steps in the right direction. (Indeed, we took a similar approach in my group many years ago, PMID: 21294912). However, the introduction as written tends to give a false impression of the progress made in this study. For example, in the introduction the authors' discuss the general problem of lack of generalizability of models. At the end of the paper, we finally get a test of one general prediction, and it's very exciting that they find some evidence for it. But in the end, it's only one prediction. So there is still a long way to go. Hence, I suggest the comments about generalizability of evolutionary models should be moved to the discussion as "future directions" for this type of research. Other examples can be found in my specific comments below.

We thank the Editor for the supportive comments and feedback that improves our manuscript. We agree with the Editor that the previous draft of the introduction was not focused and at times read more like a review/opinion piece. We made substantial changes to the introduction to make it more focused and more directly related to the work we present.

Also, there is an overall lack of reference to relevant research, especially relating to modelling of bacterial promoters/gene expression and evolution of regulatory sequences. Again, specific examples are given below.

We thank the Editor for pointing out various relevant papers, which we have now referenced throughout the manuscript.

Major comments.

AE1 1. "We found that several key evolutionary³¹ properties - the distribution of phenotypic and fitness effects of mutations, the³² evolutionary trajectories during selection for regulation - can be accurately captured³³ without accounting for all, or even most, parameters of the system"

This is one of the most interesting results/claims of the paper, but I have two comments. First, the authors should discuss is to what extent these results depend on the assumption of a quadratic fitness landscape.

The referee is absolutely correct in pointing out that our results depend on the quadratic fitness landscape assumption. We have no experimental access to the actual, environment-dependent fitness function of our system and, moreover, our system is a synthetically simplified version of the natural lambda switch. That is an inherent limitation of our work – it required us to pick a particular mathematical model for the fitness in lieu of actual experimental measurements. An assumption that we made was to pick a commonly used, quadratic model that stabilizes phenotypes at their WT values that correspond to the single peak in the phenotype space. This model has four very attractive features: first, phenotypes with highest fitness correspond, by construction, to a functional regulated promoter (high ON state, low OFF state, WT dynamics) that can be realized, both in the model and experimentally; second, this model is a theoretically well-understood approximation for generic single-peaked fitness functions close to their fitness optimum (although we also simulate far away from the peak, not only in the vicinity); third, the same mathematical form of the fitness function generalizes to phenotypes of different dimensions, allowing us to compare (eg in Figure 5E) how the dimensionality of the phenotype affects our conclusions; fourth, this choice of the fitness function naturally connects to the geometric model that we explore in Figure 6. Quadratic fitness is a minimal model with these properties, in the sense that any non-trivial results in evolutionary dynamics (e.g., possible multiple fitness peaks in the genotype space, sign epistasis, etc.) must be a consequence solely of the genotype-to-phenotype map rather than of the fitness function defined over our phenotypes. In other words, while modeling more complicated fitness functions would certainly be possible, such models would force extra non-trivial structure into evolutionary dynamics for which we have no justification in experiment or published literature.

In sum, while we believe that there are good modeling justifications for our choice, we acknowledge that that quadratic fitness indeed is our choice and not an empirical fact.

We now clearly spell out our motivation for the specific choice of the fitness function in the main text. We speculate that qualitative results that we report would not change with other fitness functions that postulate a single peak in the phenotype space, but are explicit that this is an assumption of our approach (L1027-1048, as well as other places in the manuscript).

AE2 Second, I found it extremely difficult to understand what exactly was the evidence that supports this claim. I believe that the analysis is reported in Figure 6, and the authors write:

"to accurately model evolution it is sufficient to know only704 the range of values within the EMs, but not the actual distribution of values"

However, immediately after this, they write:

"The ability to simplify systems in this manner is contingent on drawing708 the effects of mutations from the correct distribution"

which seems to contradict the previous statement. How can we draw the effects of mutations from the correct distribution without knowing the actual distribution? Perhaps these are two different distributions?

In this portion of our work, there are three possible distributions to draw mutational effects from: (i) using all the parameters from the model; (ii) geometric model that draws phenotypic effects directly; (iii) geometric model that selects energies randomly and then converts those energies to phenotypes. (i) and (iii) give similar descriptions of evolutionary trajectories, in contrast to (ii). The major difference between (i) and (iii) is that (iii) selects the effects of mutations on binding energies from a pre-defined range that corresponds to the range of energies found in the corresponding

Energy Matrices of RNAP and CI, and as such is ignorant to the internal structure and distribution of energy values within the matrix. In other words, it reduces the number of parameters from $4 \times L$ (where L is the length of the binding site/EM) to two. So, the system can be simplified if we draw mutational effects from the distribution of *energies*, with a pre-defined range. In contrast, drawing the mutational effects from the distribution of *phenotypes* (example (ii), which reduces complexity from $4 \times L$ parameters to 6) does not capture evolutionary trajectories in the same manner and hence this distribution cannot be used to simplify the system. We made several changes to the manuscript to better explain the above (L789-792; L802-803; L816-818).

We agree with the Reviewer that the term 'distribution' was confusingly used here, and we changed the text to better reflect our intentions.

AE3 They conclude:

"Accurately capturing the evolution of bacterial promoters requires accounting for the effects of mutations on binding energies, rather than directly assuming their phenotypes"

I cannot figure out what this means. It doesn't seem consistent with the claim about the simplification of the model (earlier in this section). I strongly encourage the authors to connect their conclusions specifically to what they actually found (e.g., "Accurately capturing the evolution": as far as I can tell, no measurements of accuracy are reported, capture of evolution is a weak analogy at best.)

We made several changes throughout the manuscript to better describe what we meant by 'accurately capturing evolution'. In short, we considered the 'full model' (i.e., the most complex version of the model with the greatest number of parameters we used) to be the best reflection of the reality of promoter evolution. We make this assumption but also fully acknowledge in text that this is not an accurate assumption. Then, we considered various simplified version of this most complex model and compared their performance. We made several changes to the main text to better explain our logic and not over-state our conclusions (L168-172; L809-813; L820). We also introduced two new panels into figure 6D that further demonstrate the difference in the performance of the two geometric models.

AE4 Figure 6 shows the results of complicated experiments, but it is not clearly stated how much of the variance in the DFEs/evolutionary trajectories is explained by the simpler model (152 vs. 2 parameters)? This effect should be quantified.

In the case of Fig.6D, we felt that the differences in how the two simplified models predicted promoter evolution compared to the full model were obvious, which is why we didn't quantify them. We now include a statistical comparison (Supplementary Figure 7) that provides support for our claims.

2. Overly general comments/lack of citations

AE5 -The authors study evolution of promoters from random sequences, but omit citations to studies of evolution of regulatory sequences from random. Going back at least to Stone & Wray 2001 (PMID: 11504856) and many subsequent papers. Clearly the work here is an advance, but the previous work should be discussed.

We thank the Editor for suggesting the Stone & Wray reference, which we have now included in the manuscript along with several other key references that inspired and are related to our work, both from bacterial and eukaryotic side of things.

AE6 -"On the one hand, molecular causes of evolutionary adaptations are frequently characterized, with numerous studies identifying specific mutations that underpin a given phenotypic or fitness change observed in

nature or in the laboratory. On the other hand, evolutionary biologists focus on the57 global properties of complex systems and their consequences for evolution, especially58 at the population and species levels."

I think I get the point, but we need to have specific examples of studies of both types. I also think these comments can be moved to the discussion, as they aren't really needed to introduce the current work.

We removed this section from the introduction and re-worked the introduction to be more in line with what the Editor suggested here and in other comments.

AE7 -"A similar split can be observed when considering models of evolution, which62 tend to either be abstract and general but lacking a direct connection to underlying63 molecular and cellular mechanisms; or are built to describe a specific system, but do64 not readily generalize beyond it and the experimental measurements they were65 designed to explain"

I think these comments are too general. We need specific citations of the types of models of evolution that the authors have in mind. For example, readers of GENETICS will likely know about continuous time Markov models of molecular evolution used for DNA and proteins that have no difficulty generalizing to very large numbers of sequences.

We removed this section from the introduction to improve flow and focus more specifically on the key arguments we want to make.

AE8 Also, the authors do great work to parameterize their model based on experimental measurements. However, in what sense can their model generalize beyond these measurements? In the end, there's only one qualitative prediction made. I think these comments should be moved to the discussion and can be considered as "future directions".

We agree with the Editor that the previous version of the introduction set up our work to seemingly be about generalizability of the models, when that is a relatively minor point we actually make. We hope the re-worked version of the introduction fixes this. We also made some changes to the discussion to better discuss how our work can be generalized.

AE9 -"In addition, following environmental change that affects the concentration of the TF126 in the cell, regulated promoters go through a transient phase before reaching a new127 steady-state expression level (Longo and Hasty 2006; Yosef and Regev 2011)"

The authors should add current references, especially this one seems relevant here: Cis-regulatory variants affect gene expression dynamics in yeast. PMID: 34369376

We thank the Editor for suggesting to add this reference.

AE10 -"The main aim of this work is to provide a primer on how to study biological systems of133 increasing complexity in order to understand their evolution"

I don't think "primer" is the right word. The authors use it many times in the paper, but after reading the paper, I am not sure what I have been "primed" for.

Our primary objective in writing this work was for it to serve as a guide to how mechanistic models can be used to tackle one of the major challenges in studying evolution –biological complexity. To this end, we used an example of a somewhat complex system (dynamics of gene expression from a single promoter) less so to learn specific lessons about that system (although we do that as well) and more so to demonstrate what kinds of questions could be asked through mechanistic models. It is in this sense that we see our work as a 'primer', as an 'introductory book/text on a subject' (this being

one of the definitions of the word). In spite of the Editor's comment, we have opted to keep the word in the text however will reconsider and change if the Editor does not agree with its use after reading our response. We also made substantial changes to the introduction (in response to other comments by the Editor) to clarify better the main goal of our work.

AE11 -"that function among 544 random sequences is vanishingly rare. "

De Boer's papers challenge this (at least for yeast). Worth citing here.
We added the reference as suggested.

AE12 -" utilizing a mechanistic model to remain tied to the actual while enabling the 830 exploration of the possible"

My group did something similar in our paper: "Modeling the evolution of a classic genetic switch" PMID: 21294912
We changed this sentence in the discussion to better reflect some key previous works.

AE13 -"extending the findings based on the modelling of Lambda PR promoter to other 854 regulated promoters"

Certainly, this approach can be applied to other promoters. In our paper (PMID: 21294912) we also tried to predict the genotype-phenotype relationship in a simple regulatory system using a Shea & Ackers framework, summarized the phenotype using a few dimensions and then studied evolution of the network.

I think our work should be discussed.
We added the reference as suggested.

Minor essential comments

AE14 1. "The mass action kinetics (MAK) uses standard ODEs to describe the temporal 240 dynamics of molecular concentrations in a system. In our system, MAK accounts for 241 the changes in concentrations of the repressor CI and the measurable system output, 242 YFP. "

The authors should at the very least discuss the influential literature relating to stochasticity in this promoter (and gene expression more generally). This goes back to at least Arkin & McAdams 1998 (PMID: 9691025), but has been considered many times since (PMID: 15790856, PMID: 19098103, PMID: 21119634). Some arguments about why they believe they can study evolution of this system using a deterministic approximation would be beneficial here.

We agree with the Editor that our model is deterministic and does not account for gene expression noise – neither in a temporal nor a between-individual sense. We now highlight this deficiency in the model on several occasions in the manuscript, as well as discussing how we focus on the dynamics of gene expression at the population rather than single cell level. We also now do a better job clarifying that we consider the full model (the model that includes all the parameters) as 'capturing promoter evolution' not because we think this is true (and discuss the shortcomings explicitly) but as a benchmark against which we can evaluate performance of less complex models (L168-172; L218-220; L314-317; L1001-1004).

AE15 2. "When we fitted, rather than predicted, steady-state 263

expression levels, the model fit dramatically improved for all mutants (Fig.2C,D). Our model therefore accurately predicted the effects of Lambda PR promoter mutations on the dynamics of gene expression, with the error largely stemming from comparatively poorer predictions of steady-state expression levels"

This argument is not correct from a statistical perspective unless the authors know that they are fitting the "true" model and the parameter estimates are independent. Since many features of gene expression have been approximated and the model is complex, these assumptions are unlikely to be true. (Consider fitting a line to a polynomial. Surely you can improve the fit of the line by adjusting the intercept. But this does not imply that the error results only from an incorrect guess of this parameter.)

We changed the text in this section to reflect Editor's comment (L304-306; L312).

AE16 3. "In other words, some changes to gene expression phenotypes cannot be achieved by mutating the promoter and might have to rely on changes to other components of gene regulation machinery (such as TFs)"

As far as I understand it, the authors only considered single and double mutations in the promoter. To make this claim, they would need to rule out indels, duplications, horizontal gene transfer, etc. The claim is unnecessary and can just be cut.

Deleted as suggested.

AE17 4. "This finding suggests that the internal structure of the EMs might have been selected for."

"Put together, the constraints acting on genotype-phenotype mapping arise from varied aspects of the system"

"to describe the fitness of a given random sequence, we assumed a quadratic fitness landscape (a common model of fitness effects in evolutionary biology), with the phenotypic values of the wild type Lambda PR defining the optimum."

These comments seem too much. The authors are considering only a single promoter in isolation. We have no idea to what extent this reflects the biological function of this promoter in the context of the intact regulatory network. In other words, the authors are considering gene expression dynamics of a single promoter as a "phenotype" but have not actually connected it to fitness. The assumption of a quadratic fitness function seems ok for phenotypes very near the wt, but is probably not very good for phenotypes far from the wt.

We made changes to these sections to reflect Editor's comments (L543; L629-639).

November 13, 2024

RE: GENETICS-2024-307483

Dr. Mato Lagator
The University of Manchester Faculty of Biology Medicine and Health
Evolution, Infection and Genomic Sciences
Oxford rd
Manchester
United Kingdom

Dear Dr. Lagator:

Congratulations! We are delighted to inform you that your manuscript entitled "Linking Molecular Mechanisms to their Evolutionary Consequences: a primer" is acceptable for publication in GENETICS. Many thanks for submitting your research to the journal.

The reviewers had a few suggestions for improving the manuscript that you may want to consider. You can view their comments at the bottom of this email.

To Proceed to Production:

1. Format your article according to GENETICS style, as discussed at <https://academic.oup.com/genetics/pages/general-instructions>, and upload your final files at <https://genetics.msubmit.net>.
2. Your manuscript will be published as-is (unedited-as submitted, reviewed, and accepted) at the GENETICS website as an Advanced Access article and deposited into PubMed shortly after receipt of source files and the completed license to publish. Please notify sourcefiles@thegsajournals.org if you do not wish to publish your article via Advanced Access.
3. We invite you to submit an original color figure related to your paper for consideration as cover art. Please email your submission to the editorial office or upload it with your final files. You can submit a small-sized image for evaluation, and if selected, the final image must be a TIFF file 2513px wide by 3263px high (8.375 by 10.875 inches; resolution of 600ppi). Please avoid graphs and small type.

If you have any questions or encounter any problems while uploading your accepted manuscript files, please email the editorial office at sourcefiles@thegsajournals.org.

Sincerely,

Alan Moses
Associate Editor
GENETICS

Approved by:
Audrey Gasch
Senior Editor
GENETICS

note: Please add jnls.author.support@oup.com and genetics.oup@kwglobal.com (or the domains @oup.com and @kwglobal.com) to your email program's "safe senders" list. You will be contacted by both at various points during the production process.

Review comments (if applicable):

Reviewer #1:

The authors have addressed most of my concerns. In most cases, however, they have now cited previous work, but not actually

discussed what was found previously. In future, I encourage the authors to go beyond citations and succinctly explain what was found previously and whether/how their findings support, refute or re-contextualize it.

Reviewer #2 :

The authors have satisfactorily addressed my comments and concerns from the previous review. The paper makes a significant contribution to the field.